# Life and death of colloidal bonds control the rate-dependent rheology of gels

Mohammad Nabizadeh [1✉] & Safa Jamali [1✉]

Colloidal gels exhibit rich rheological responses under flowing conditions. A clear understanding of the coupling between the kinetics of the formation/rupture of colloidal bonds and the rheological response of attractive gels is lacking. In particular, for gels under different flow regimes, the correlation between the complex rheological response, the bond kinetics, microscopic forces, and an overall micromechanistic view is missing in previous works. Here, we report the bond dynamics in short-range attractive particles, microscopically measured stresses on individual particles and the spatiotemporal evolution of the colloidal structures in different flow regimes. The interplay between interparticle attraction and hydrodynamic stresses is found to be the key to unraveling the physical underpinnings of colloidal gel rheology. Attractive stresses, mostly originating from older bonds dominate the response at low Mason number (the ratio of shearing to attractive forces) while hydrodynamic stresses tend to control the rheology at higher Mason numbers, mostly arising from short-lived bonds. Finally, we present visual mapping of particle bond numbers, their life times and their borne stresses under different flow regimes.

[1] Department of Mechanical and Industrial Engineering, Northeastern University, Boston, MA, USA. ✉email: nabizadehmashhadto.m@northeastern.edu; s.jamali@northeastern.edu

Colloidal particles that are attractive to one another in a range of state variables (solid fraction, strength, and range of interactions) self assemble into particulate networks with an overall gel-like mechanical behavior[1–8]. At relatively small fractions of colloids, continuous structures can be formed by very large attractions while at large volume fractions, gelation can occur at smaller attraction strengths. The resulting structures formed under quiescent conditions have been studied extensively with respect to these state variables[4,9–12]. At the intermediate volume fractions of particles and reversible attractions, the spatially distributed networks are found to be heterogeneous (fractal) where colloidal structures are formed at different length scales from local glassy structures at small length scales to spatial spanning fractal clusters at long range[9,13–17]. Originally, theoretical and computational investigations[13] and more recently an experimental work[14] have shown that these locally glassy structures are the origin of elasticity in colloidal gel networks.

The hallmark of these colloidal gels is the strict coupling of their macroscopic mechanical properties with both the microstructure and the flowing conditions that they are subject to[18]. This coupling has been studied within the context of soft glassy rheology[19–21]. Recently, detailed simulations of gelation under flowing conditions have shown that structures that are formed under flow significantly differ from ones that are formed in quiescent conditions, indicating a strong role to be played by the fluid flow in defining the resulting particulate structures[22]. These spatially distributed particle structures exhibit a wide range of exotic rheological responses from an emergence of a yield stress under which the material behaves similar to a viscoelastic solid, to time- and rate-dependent flows that are referred to as thixotropic elasto-visco-plastic (TEVP)[19,23–28]. Understanding the micromechanistic view of the physical underpinnings in these complex rheological behavior is essential in controlling the mechanics of colloidal gels. Recent advances in scattering techniques[29–31] as well as detailed computer simulations[22,23,32–34] have brought an invaluable insight into understanding the evolution of the colloidal structures under different flowing conditions. A focal point of many studies have been the fundamentals of yielding in colloidal gels[35]. In contrast, the dynamical behavior of these materials, and more specifically the structure–flow coupling at longer times and in the quasi-steady state remains largely unexplored.

There is a growing body of literature now suggesting that it is the competition between the interparticle attractive forces and the hydrodynamic shearing forces that determine the state of the particulate microstructure and hence the mechanical response of a colloidal gel[14,22,23,31,34,36–39]. This competition can be understood using a dimensionless group referred to as the Mason number, ($Mn = 6\pi\eta_0\dot{\gamma}a^3/U_0$), in which the $\eta_0$, $\dot{\gamma}$, $a$, and $U_0$ are solvent viscosity, applied deformation rate, particle radius, and the strength of attractive potential between the particles. At small $Mn < 0.01$, the shearing flows do not result in fluidization of the particulate structure and in contrast lead to shear compaction of the structure. At intermediate $0.01 < Mn < 1$ the consistent breakage and reformation of particle–particle bonds result in structural and kinematic heterogeneities to emerge, and ultimately at large $Mn > 1$, the shearing forces effectively break the particle bonds resulting in a fluid-like flow similar to a suspension of non-interacting colloids[23]. Mason number, first introduced for electrorheological fluids[40,41] is used throughout this study. However, a Buckingham theorem dimensional analysis of similar force comparison was proposed by Xie et al.[42], suggesting a Breakage number, Br. Additionally, in a series of previous studies Peclet of depletion, $Pe_{dep}$, was introduced by Koumakis and coworkers[27,43] and later used by others[37–39]. While these dimensionless groups are all similar in essence, we use the Mason number as the dimensionless group of choice as it provides a foundation to compare the magnitude of two competing forces in the system.

On the other hand, many studies have shown that there is a relationship between the macroscopic mechanics of the system and the particulate mesostructures that are formed when the system is driven far from equilibrium[43,44]. Recently, Rocklin et al.[45] proposed a simple micromechanical model, describing the moduli of a colloidal structure using the particle–particle bond information.

The seminal work of Evans and Ritchie[46] showed by obtaining a distribution of bond lifetimes in self-associating systems under applied deformations and their rupture events, an energy landscape for the system far from equilibrium can be described. For the gelation process, Coniglio et al.[47] suggested a correlation between the lifetime of bonds and the gel microstructure using a cut-off limited cluster distribution approach, showing that the average lifetime of bonds within a cluster grows with the formation of larger clusters. In dense short-ranged attractive colloidal glasses, finite bond lifetime leads to glass transition and is believed to be the main source of elasticity[48]. The rheological response of colloidal glasses is also shown to be directly originated from rearrangements of long-lived neighbors[49]. Additionally, the hydrodynamic interactions significantly prolong the lifetime of particle bonds[50]. On the other hand, particle bond lifetimes under quiescent conditions are believed to be strong functions of temperature and interparticle interactions rather than the solid volume fraction[8]. What is clear is that the particle bonds and their coupling to the flow conditions have a profound impact on the response of the particulate ensemble to an applied deformation/stress. However, the dynamical strength of these bonds under different deformation rates is yet to be understood. It is also not clear which interactions are dominant at different flow conditions, and what are their implications on the colloidal structures and their lifetimes.

In this work, we employ a detailed simulation of large particle ensembles that accurately preserve the essential hydrodynamics of the system[37,51,52] to study the evolution of the microstructural characteristics of the system such as bond distribution, load-bearing, and lifetime. Having access to all spatiotemporal configurations of colloids, we study the lifetime of particle–particle bonds formed under different conditions and their correlation to stresses exerted on those particles, and the bulk response of the colloidal system. We find that the dynamics of the short-lived bonds are determined by the hydrodynamics, and as the bonds get older, their dynamics become more attraction-dominated. We also observed that when sheared, similar to the bulk response, a stress overshoot is observed for microscopically measured stresses of bonds during shearing, interestingly, there scaling of the stress overshoot with strain is similar for the micro- and macro-scale.

## Results

**Decoupling stresses and visual inspection.** Colloidal gels are thixotropic in nature, with long time (compared to bare particle diffusion timescale) kinematic and structural heterogeneities under intermediate Mason numbers[22,53]. Thus, here we limit our study to the quasi-steady state response of the colloidal system to an applied deformation rate, with respect to the measured shear stress. The colloidal gel formed under quiescent conditions is modeled initially for ~500 diffusion times to assume a quasi-steady microstructure, followed by imposing a constant deformation rate flow protocol. The time evolution of the stress response is then monitored and when reached the quasi-steady state reported in Fig. 1a, against the imposed Mason number. The gels' shear stress response clearly indicate an emergence of a yield stress, followed by a significant shear-thinning regime, and

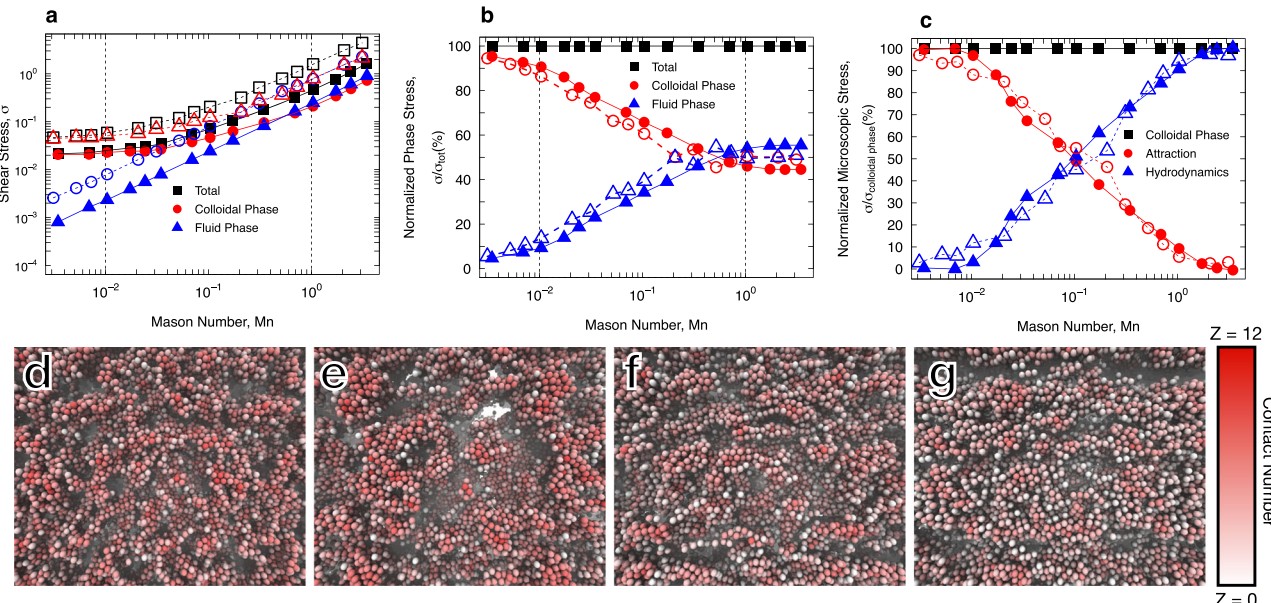

**Fig. 1 Decomposed shear stresses, different flow regimes, and microstructure under flow. a** Shear stresses plotted against Mason number, Mn, in the form of fluid, colloidal, and total stresses, **b** the relative contribution of each phase's stresses to the total stress, and **c** the relative contribution of the attractive and hydrodynamic microscopic forces to the macroscopic stress. The filled symbols and solid lines correspond to attraction strength of $U_0 = 6k_BT$, and open symbols with dashed lines correspond to attraction strength of $U_0 = 20k_BT$ between colloids. Snapshots of the colloidal particles, color-coded with contact number at **d** quiescent condition, Mn = 0.0, and the shear-induced structuration regime: **e** Mn = 0.03, **f** Mn = 0.24 and, **g** Mn = 0.7. All snapshots correspond to the attraction strength of $6k_BT$.

ultimately a pseudo-Newtonian viscosity at large Mason numbers. Nonetheless, the fluid phase shows a Newtonian behavior, i.e., a linear increase in the shear stress response with deformation rate. The colloidal stress on the other hand shows an evident yielding behavior, suggesting that while at low deformation rates, the colloidal phase's response determines the overall mechanics of the system, at high Mn regime increasingly becomes screened by the fluid's response. This is clear in Fig. 1b, presented as the fraction of overall stress response borne in each phase. Since the complex rheological response of the ensemble is controlled by the particulate phase, one can further decouple the source of stress in the material according to different microscopic forces in the colloidal particles. The colloidal dynamics in short-range attractive systems are governed by the sum of two different types of interactions: (i) Hydrodynamic, and (ii) interparticle (attractive) interactions. Note that the Brownian forces are random in nature and significantly weaker than the attractions studied in our work, and hence can be ignored. In our scheme, the hydrodynamic interactions are measured through colloid-fluid interactions and dissipative interactions that represent the long-range hydrodynamic, as well as the short-range lubrication between interacting colloidal particles. The fraction of stresses originated from hydrodynamic and attractive interactions are plotted in Fig. 1c against Mason numbers. First of all, the results in Fig. 1c provide a micromechanistic description to previously suggested classification of three (3) main Mn regimes[23]: at (Mn < 0.01), shear-compaction regime, the attractive interactions give rise to large stress responses followed by a large shear-thinning regime, in 0.01 < Mn < 1, the shear-rejuvenation regime, the competition between hydrodynamic and attractive forces gives rise to long time transient and dynamic behavior, and ultimately at Mn > 1, the fluidization regime, the strong hydrodynamic interactions, screen interparticle interactions. In the first regime, the colloidal network is rather unchanged with slight coarsening and compaction over time, and in the fluidization regime, the network is nonexistent as the colloidal bonds are effectively broken. In the

second regime secondary flow-induced structures emerge that are entirely different from the initial gel structures, and thus we refer to this regime as shear rejuvenation. It should be noted that some literature refers to shear rejuvenation when a high rate of shear is applied to remove any previous memory, followed by cessation of flow to re-construct the gel microstructure. Results in Fig. 1a–c also clearly show that when presented against Mason number, the total shear stress, as well as individual contributions from the colloidal phase (both attractive and hydrodynamic forces), follow virtually identical trends for different strengths of attraction. While the yield stress of the colloidal network increases for higher attraction strengths, the source of stresses and their relative contribution remain unchanged, further suggesting that the Mason number appropriately classifies the rheological behavior of these materials.

The snapshots of the system's initial microstructure, as well as ones under three different deformation rates in the intermediate Mn regimes, are presented in Fig. 1d–g. The snapshots here clearly show that in this shear-induced structuration regime, secondary clusters and aggregates form that vary in size and orientation as a result of competition between attractive and hydrodynamic forces. Colloidal particles are color-coded with a warmer (red) color corresponding to a larger number of contacts. Note that in contrary to the experimental definition of particle–particle contact, which commonly constitutes a criterion based on the radial distribution function of particulate phase and is associated with intrinsic imaging uncertainties, here we identify two particles in contact when the attractive force acting between them exceeds the thermal/Brownian forces. Using the parameters previously outlined, this definition results in colloids with $h < 0.1a$ to be identified as contacting neighbors, where $h$ measures the surface-surface distance between two interacting colloidal particles.

**Coordination number and load bearing of particles.** The coordination number of colloidal particles and their distributions

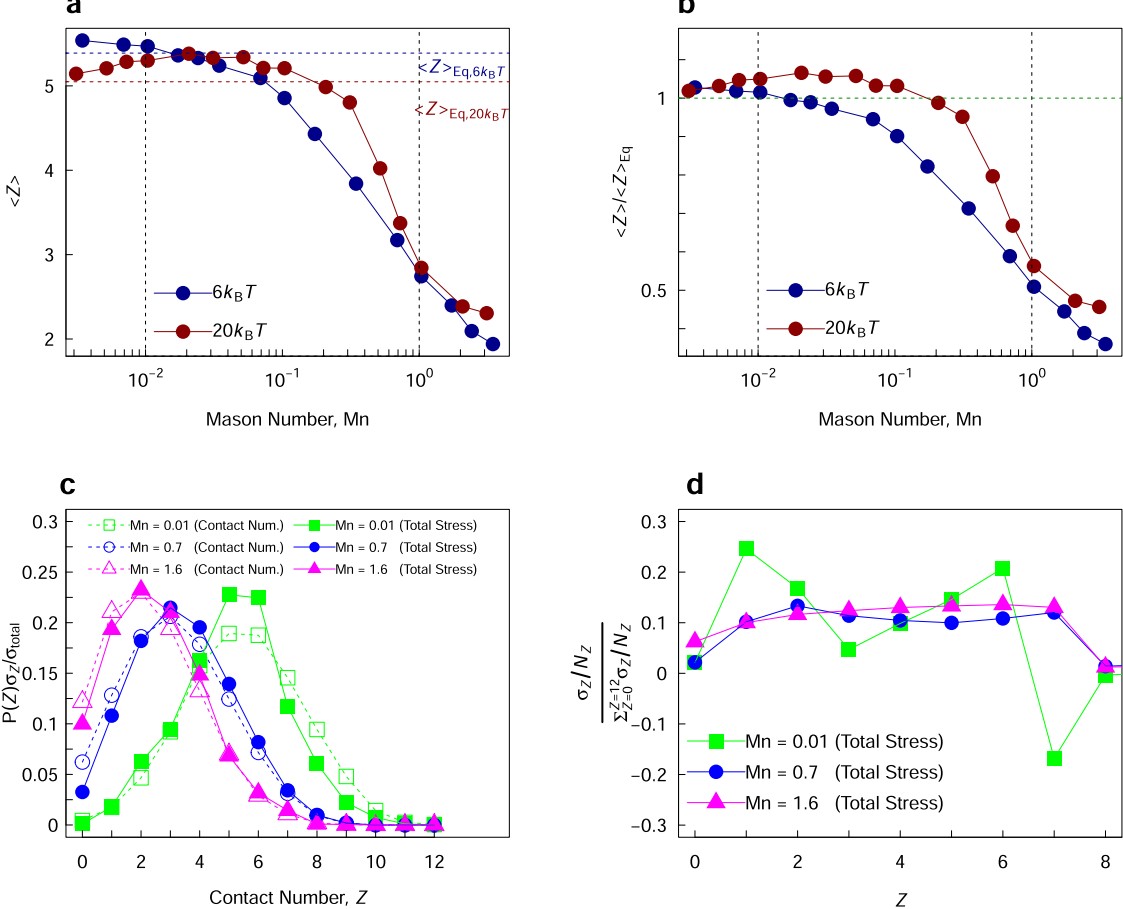

**Fig. 2 Microstructure evolution and distributions in different flow regimes. a** Average contact number at each Mason number at the quasi-steady state. Values are presented for two different attractions strengths of $U_O = 6, 20 k_B T$, with the initial value of the average coordination number marked with a dashed horizontal line. **b** Normalized average coordination number by its initial value [of a quiescent gel]. **c** Distribution of contact numbers (dashed line) and the relative contribution of stresses borne on each contact population (solid lines) against coordination number for different imposed deformation rates. And **d** the contribution of stress per particle, for each coordination population ($U_O = 6 k_B T$).

are quantified and represented in Fig. 2. The average coordination numbers, <Z> as a function of applied Mason number, plotted in Fig. 2a, is in agreement with the previous experimental measurement of[31]. In the first regime, <Z> remains virtually intact, as the dominating attractive forces effectively keep particles together and retain the structural integrity of particulate clusters. The dynamic and consistent breakage and formation of particle bonds result in a non-monotonic (or monotonic depending on the attraction strength) decrease of <Z> in the intermediate Mn regime, followed by a terminal regime where <Z> can essentially be predicted using the collisional dynamics of particle population at each volume fraction. Furthermore, the comparison between the average number of bonds per particle under flow shows that while there are quantitative differences in the values measured for weak and $U_O = 20 k_B T$, the trends when presented against the Mason number follow a universal behavior, as previously shown and discussed in ref. [37]. In the initial shear-compaction regime, the average value of the coordination number for the entire system grows slightly above its measure at the quiescent conditions, as indicated in Fig. 2b. However, the extent of this increase depends upon the volume fraction of particles, and is more visible for the lower fraction of colloids in the system.

The distribution of contact numbers in Fig. 2c shows an evident shift to smaller values with the increase of imposed deformation rate. Furthermore, the contribution of each contact number population follows the distribution of those particles very

closely. In another word, while most of the total colloidal stresses at the Mn = 0.7 are borne in particles with three neighbors, $Z = 3$, it is simply owed to their highest population amongst all coordination numbers. This is clearly evident in Fig. 2d, where the contribution to the total stress per particle is plotted for different Mn values. Thus, the macroscopic stresses in the system cannot be directly correlated to a single contact population/ information for the entire Mn ranges.

**Kinetics of bond rupture/formation under shear.** Having the particle bonds clearly identified in the system, we track all particles and their bond dynamics (rupture/formation events) throughout the entire flow protocol. We then measure the life-time of particle bonds for colloidal ensembles under different Mn numbers at a quasi-steady state. Since the bond lifetime is measured in DPD units, we normalize all bond lifetimes with the diffusion timescale of a bare particle ($\tau_0 \approx 5.2$, calculated from our simulations). As the particles are subject to different deformation rates, bonds are consistently formed and broken, resulting in a distribution of bond lifetimes to be present at each instance. Figure 3a presents the distribution of such lifetimes for three different Mason numbers at the end of our flow protocol and in quasi-steady-state conditions. Note that we only consider the bonds that are formed and broken during this steady-state time window. In another word, the pre-existing bond ages prior to reaching steady-state condition are not considered in these

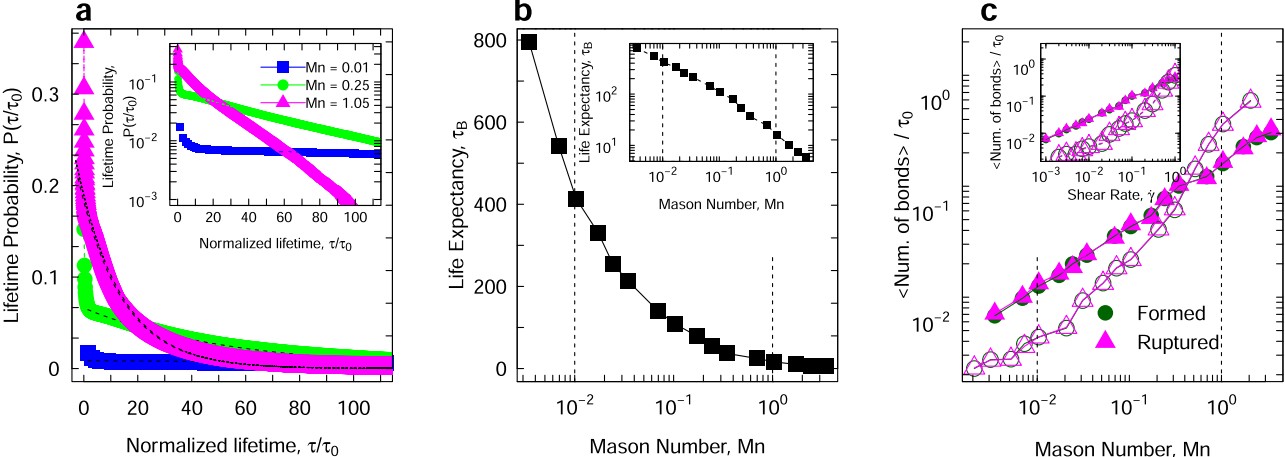

**Fig. 3 Kinetics of a colloidal bond lifetime. a** The relative distribution of bond lifetimes normalized by the diffusion time of a bare particle for different $Mn$ numbers. Dashed lines represent an exponential decay fitting function as $P(\tau) = C\exp(-\tau/\tau_B)$, where $C$ is a conversion factor constant. The insert graph shows the same probabilities in a semi-log graph. **b** the life expectancy (in $\tau_0$ units) calculated from the lifetime distributions versus Mn number, with the insert graph showing the same data in a log–log plot. and **c** the average number of formed/ruptured bonds per particle during a single diffusion time. The filled symbols and solid lines correspond to attraction strength of $U_0 = 6k_BT$, and open symbols with dashed lines correspond to attraction strength of $U_0 = 20k_BT$ between colloids. The insert shows the average number of formed/ruptured bonds per particle versus shear rate.

calculations to remove any biases that reflect on the yielding behavior of different gels. The distribution of bond lifetimes for different Mn values shows that as the deformation rate increases, the fraction of short-lived bonds dramatically increases (the population of long-lived bonds decreases). The bond survival probabilities as shown in Fig. 3a show an exponentially decaying behavior for all Mn numbers. Here, we fit a decay function as $P(\tau) = C\exp(-\tau/\tau_B)$, where $C$ is a conversion factor constant, $\tau$ is bond lifetime and we name $\tau_B$ as the life expectancy. We define this characteristic timescale, $\tau_B$, as the bond life expectancy. The bond life expectancy against the applied Mn is presented in Fig. 3b, showing a power-law dependence on the imposed deformation rate. The decrease in the life expectancy suggests a fast breakage/formation dynamic for the colloidal bonds, and does not necessarily imply a lower number of bonds. Also to be noted is the fact that these bond dynamics are measured at the quasi-steady state and far from yielding dynamics, where the rate of bond formations and bond breakages take different values as suggested by Colombo et al.[32]. The consistent and steady bond numbers can be verified in Fig. 3c, showing exactly the same rate of broken and formed bonds at different Mn values. Note that the results in Fig. 3b are measured from the characteristic relaxation time of lifetime distribution probabilities in Fig. 3a, while the rates of ruptured and formed bonds are measured directly from the simulations and monitoring the number of bonds per diffusion time. In the shear-compaction regime, each particle loses ~0.002$Z$ of its contacts during each diffusion time, in stark contrast to more than 0.05$Z$ in the fluidization regime (Fig. 3c). Nonetheless, this average rate representation does not mean that long-lived bonds do not exist at high Mason numbers. Furthermore, the results for different attractions strengths show that while the rate of formed/broken bonds changes for different values of attraction, the behavior shows a similar trend when plotted against the Mason number. The results in Fig. 3c, and alternatively the bond lifetimes, can be also compared to the theoretical framework based on Kramer's escape rate model[54,55]. Based on this theory, rate of aggregate breakage shows a power-law dependence on the applied deformation rate, transitioning to an exponential function in a shear-controlled regime. While such calculations do not include the complex many-body dynamics and their

resulting hydrodynamics on the bond breakage and formation for the interaction-dominated regime. Our results, based on total rates of bond breakage and formation in the quasi-steady state regime qualitatively agree with the theoretical predictions. In Fig. 3c, the rupture/formation rate of bonds for both the strong ($U_0 = 20k_BT$) and weak ($U_0 = 6k_BT$) gels depend on the applied Mason number through a power-law function. The contribution of hydrodynamic stresses becomes comparable to the ones from attractive forces at Mn = 0.2 (Fig. 1c), coinciding with the point where rupture rates of the two gels become comparable (Fig. 3c). For higher Mn, the rupture rate of bonds for the stronger gel ($U_0 = 20k_BT$), becomes more frequent than that of the weak gel ($U_0 = 6k_BT$). This is expected, as a higher shear rate is required for the stronger gel, to achieve the same Mn, compared to the weak gel. Thus at high deformation rates, the bond dynamics are controlled/described through shear rate rather than the Mason number (inset of Fig. 3c shows the average bond rupture/formation frequency of the two gels fall on top of each other when plotted against shear rate). In this regime, since shear flow effectively breaks down the majority of colloidal bonds, these rates are controlled by the collisional dynamics and the relative velocity of particles, i.e., shear rate; however, at lower deformation rates and since colloidal bonds resist motion in different planes of shear, Mason number becomes the key indicator of these dynamics.

By correlating results in Figs. 1c and 3b, one can clearly derive that for systems with a small average lifetime, hydrodynamic stresses are dominant and attractive forces are negligible whereas, for systems with large average bond life, attractive forces are dominant. There is also a clear transition between the two where hydrodynamics and attraction both play an important role in the stress of the colloidal phase.

For systems with $\tau_B > 400\tau_0$, attractive forces are the primary source of macroscopic stress and by dominating the hydrodynamic forces effectively increase the bond life expectancy. On the contrary, short-lived neighbors with $\tau_B < 4\tau_0$ are controlled by the hydrodynamic forces. Bonds with the intermediate life expectancy of $6\tau_0 < \tau_B < 400\tau_0$ are however controlled through an interplay between the hydrodynamics and attractive forces. Now and with the dynamical bond lifetime information as well as the macroscopic stresses at hand, we can identify the particle

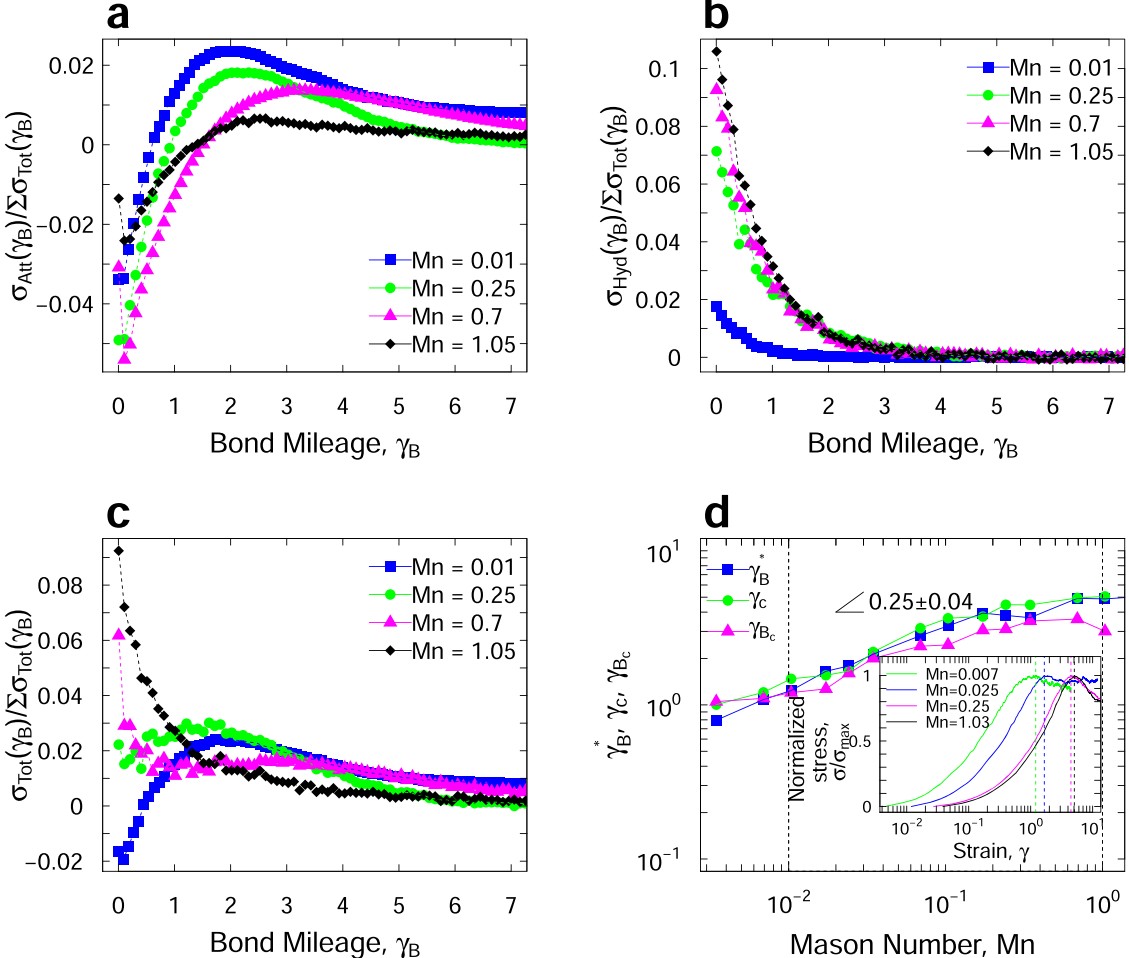

**Fig. 4 Microscopic stresses, bond mileage, and critical strain scaling. a** The contribution of the attractive forces, $\sigma_{Att}(\gamma_B)$, **b** the contribution of hydrodynamic forces, $\sigma_{Hyd}(\gamma_B)$, and **c** the total stress, $\sigma_{Tot}(\gamma_B)$, measured for different bond populations, represented as the total bond mileage, and normalized by the total stress measured over all bond mileages in the system for different Mason numbers. **d** The characteristic bond mileage from life expectancy, ($\gamma_B^* = \tau_0\dot{\gamma}$), the critical strain for stress overshoot under flow start-up experiments ($\gamma_c$), and the bond mileage with the maximum attractive stresses ($\gamma_{B_c}$) against the imposed $Mn$ number. The insert graph shows the stress against strain curves upon inception of shear flow, from which the critical strain for stress overshoot is measured for a few sampled Mason values.

bonds with respect to their contribution to the total macroscopic stresses in the material under flowing conditions. The rheology of colloidal gels is not only rate-dependent, but also time-dependent, as they exhibit a wide range of thixotropic behavior under different flow regimes. For instance, the final mechanical response of the suspension, when brought to rest from the flowing condition, can be described based upon the total strain that the particles have experienced[22]. Additionally, the fabric of the particulate network is found to exhibit a universal behavior with respect to its shear and deviatoric components, when presented against the shear strain instead of shear rate or Mason number[23]. While the life expectancy identifies the average lifetime of a colloidal bond under an applied deformation, a strain measure of the populations that have experienced the same shear history would be beneficial. Accordingly, we define the total accumulated strain observed over the lifetime of each bond, as its bond mileage, $\gamma_B$.

**Dynamics of bonds and critical strain scaling.** Figure 4a, b shows the contributions of attractive and hydrodynamic forces to the total stress of the colloidal phase against the bond mileage at different $Mn$ values, respectively. The bond mileages presented here are sampled over the entire range of quasi-steady-state flow.

To do this, all populations of bond lifetimes and their corresponding stresses are tracked and data are collected over the entire flow protocol. Note that although bond mileage is of the same nature as shear strain, it reflects on a population of bonds that share the same total number of flow strains, regardless of the point they were formed. Thus, it is important to differentiate between the bond mileage and shear strain, as one refers to all bonds that are formed [at different times] and ruptured after a certain number of strains, and the other measures the flow strains.

First of all, the results in Fig. 4a clearly indicate that regardless of the bond age, all young bonds contribute negatively to the total attractive stresses, transitioning to positive contributions at bond mileages of $\gamma_B > \sim1$ accumulated strains. This is followed by a clear maximum for middle-aged bonds with mileages $1 < \gamma_B < 4$ for different $Mn$ numbers, and ultimately a positive rate-independent attractive stress contribution for the old bonds. On the other hand, Fig. 4b suggests that while all bonds contribute positively to the hydrodynamic stresses in the system, this contribution becomes relatively negligible for bond mileages of $\gamma_B > 2$ strains. In the first and second regimes, $Mn < 1$, all of the overall contributions of bonds with different mileages to the macroscopic stress in Fig. 4c go through a maximum, while at

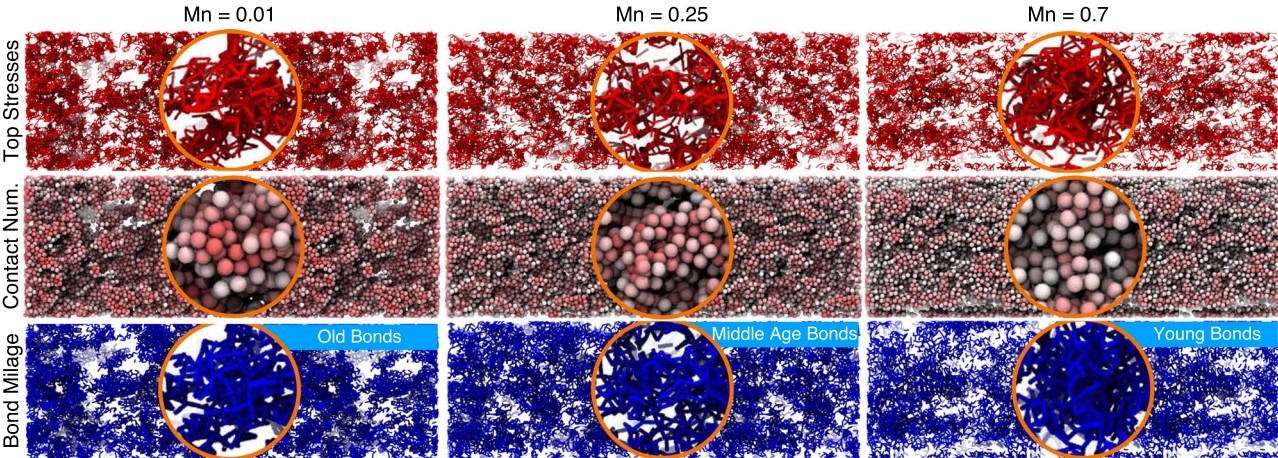

**Fig. 5 Visualization of colloidal bond ages, stresses, and contact numbers.** Quasi-steady state snapshots in three different Mason numbers (the three columns), where the first row of represents the bonds that rank >95% in load bearing, the second row only shows particles, color coded by their coordination number (increasing from white to red), and in the third row, we only show age-dependent population of bonds (old for Mn = 0.01, middle-aged for Mn = 0.25 and young for Mn = 0.7).

Mn > 1, the macroscopic stress monotonically decreases with the increase of bond mileage. Note that as shown in Fig. 1, in the fluidization regime the mechanics are determined by the fluid phase and thus these bond stresses do not control the overall response of the material. The peak bond mileage in this graph indicates the bonds that on average bear the largest macroscopic stresses measured at each deformation rate. This characteristic bond mileage shown in Fig. 4d increases by increasing the applied Mn, scaling as $\gamma_B^c \propto Mn^{0.25}$. This critical bond mileage coincides with the characteristic bond mileage of the system calculated from the life expectancy at each Mn number, $\gamma_B^* = \tau_B \times \dot{\gamma}$. The time-dependent stress response of colloidal gels to an applied deformation rate upon inception of flow undergoes an overshoot at a critical strain, which increases by increasing the rate of deformation. The inset graph in Fig. 4d shows the stress overshoot at the initial shear start-up flow. our results suggest that these critical strains at which stress overshoot is observed, $\gamma_C$, also show the same trend and scaling as the maximum in the stress-bond mileage graphs. This is significant, since the bond mileages are all calculated in the quasi-steady state and far from this initial departure from the linear viscoelastic response, and yet reflect on the microstructural and dynamical sources of the stress overshoot. It's worth mentioning that our simulations predict higher strain values at the stress overshoot compared to other works[32,37–39,43], while the same scaling is observed. This difference arises from the dependency of the critical strain on the simulation parameters and the shearing protocol, therefore, our discussion is solely on the scaling behavior for the critical strain at stress overshoot, $\gamma_C$, the critical bond mileage at attractive stress overshoot, $\gamma_{BC}$, and the critical expected bond mileage, $\gamma_B^*$. The critical strains are directly calculated from the initial flow simulations, where the stress response at the inception of shear flow is monitored. Previous computational and experimental efforts[23,27,32,37,39,43] have reported in detail, different mechanisms of yielding including a single and two-step yielding behavior in colloidal gels, also referred to the static and the dynamic yielding. While the two-step yielding and emergence of two different overshoots are not uncommon, here we have only focused on the first stress overshoot. The strain at which this overshoot is measured is reported to have different trends both in experiments and computation, but here for the sake of consistency we are measuring these strains from our flow start-up simulations.

Our results suggest that the magnitude of the hydrodynamic force plays a central role in the yielding of the structure in attractive colloidal systems. It should be noted that the Mason number alone provides a particle-level force scaling, and thus is not an accurate measure of the actual forces acting on the individual, clusters, or networks of particles. As such, the screening of the hydrodynamic interactions for particles buried within the clusters, many-body effects, and collective motion of colloidal aggregates will profoundly change the actual forces acting on different particles. Our study, leveraging the pairwise nature of interactions introduced, measures and decouples the overall acting forces on ensembles of particles. Knowing the life expectancy of bonds at different Mason numbers, and the average stress contribution of each bond mileage, a complete picture of the sources of macroscopic stress under flow can be drawn: (i) In the shear-compaction regime, Mn < 0.01, long-lived neighbors/old bonds dominated by the attractive forces govern the macroscopic response of the material, (ii) In shear-induced structuration regime, 0.01 < Mn < 1, middle-aged bonds with a meaningful competition between the attractive and the hydrodynamic forces are responsible for the overall stress, and (iii) At large deformation rates in the fluidization regime, Mn > 1, hydrodynamically dominated short-lived neighbors/young bonds control the macroscopic mechanics of the solid phase, while the overall system is dominated by the fluid phase response. These can be distinctly observed in the snapshots of the system, visualized with respect to bond life, coordination numbers, and borne stresses in Fig. 5. The stresses visualized in Fig. 5 represent the bonds with the top 5% of the shear stresses in the system (80–95% top stresses also shown through thinner lines). By comparing the snapshots of visualized stresses to the <Z>-coded particle networks, one could argue the absence of any correlation between the particle coordination number and the stresses. On the other hand, representing the young, middle-aged, and old bonds as the Mn value increases clearly shows a direct correlation between the bond lifetime and the macroscopic stresses.

## Discussion
Our simulations reveal the connection between the mileage of a colloidal bond under shear, and the role that bonds play in the resulting rheology of the short-ranged attractive colloidal gels. At small deformation rates, long-lived neighbors are responsible for the rheology of the gel, similar to what has been observed in

unperturbed colloidal glasses[13,49]. At small deformation rates, and in the shear-compaction regime, the colloidal network remains mainly intact. Thus one would expect that the long-lived neighbors are also responsible for the macroscopic rheology in this regime. Our results not only confirm this micromechanistic perspective, but also suggest that this is in fact a rate-dependent behavior that cannot be generalized to other deformation rates. As such, with the increase of imposed deformation rate, the rheology of attractive colloids is governed by progressively younger bonds. Our results also show that while the dynamics of young bonds/short-lived neighbors are controlled by the hydrodynamic forces at the microscale, they become dominated by attractive forces as they grow older in old bonds/long-lived neighbors. Finally, our results indicate that the stress overshoot observed in the shear stress response of colloidal gels under flow start-up experiments can be directly correlated to bond mileage in the quasi-steady state.

## Methods

**Dissipative particle dynamics (DPD) simulations.** Dissipative particle dynamics (DPD) is a discrete fluid model, where all particles including the background fluid particles are modeled explicitly and through pairwise interactions.

We write the equation of motion for the DPD formalism as:

$$m_i \frac{d\mathbf{v}_i}{dt} = \sum_{i,\, i \neq j}^{N_p} \left( \mathbf{F}_{ij}^C + \mathbf{F}_{ij}^D + \mathbf{F}_{ij}^R + \mathbf{F}_{ij}^H + \mathbf{F}_{ij}^M \right) \tag{1}$$

$$\mathbf{F}_{ij}^R = \sigma_{ij}\omega_{ij}(ij_{ij})\Theta_{ij}\Delta t^{-1/2}\mathbf{e}_{ij} \tag{2}$$

$$\mathbf{F}_{ij}^D = \gamma_{ij}\omega_{ij}^2(r_{ij})(\mathbf{v}_{ij}\cdot\mathbf{e}_{ij})\mathbf{e}_{ij} \tag{3}$$

$$\mathbf{F}_{ij}^C = a_{ij}\omega_{ij}(r_{ij})\mathbf{e}_{ij} \tag{4}$$

$$\omega_{ij} = (1 - r_{ij}/r_c) \tag{5}$$

$$\mathbf{F}_{ij}^H = \mu_{ij}^H(\mathbf{v}_{ij}\cdot\mathbf{e}_{ij})\mathbf{e}_{ij}, \tag{6}$$

In Eq. (1), $\mathbf{F}_{ij}^C$, $\mathbf{F}_{ij}^D$, $\mathbf{F}_{ij}^R$ are the pairwise conservative, dissipative, and random forces, respectively. Random and dissipative forces together form the canonical ensemble and satisfy the fluctuation–dissipation requirements. The random force Eq. (2) introduces thermal fluctuations via a random function, $\Theta_{ij}$. This heat is dissipated by the dissipative force Eq. (3) acting against the relative motion of particles $\mathbf{v}_{ij} = \mathbf{v}_{ij} - \mathbf{v}_{ij}$. $\gamma_{ij}$ is the strength of dissipation, coupled with the thermal noise, $\sigma_{ij}$. Together these parameters define the dimensionless temperature as $k_B T = \sigma_{ij}^2/2\gamma_{ij}$. $\Delta t$ is the time step used in the simulation and $\mathbf{e}_{ij}$ is the unit vector. Conservative force Eq. (4), defines the chemical identity of a particle based on its chemical potential/solubility in the system, through a parameter, $a_{ij}$. The random, dissipative and conservative forces are calculated via a weight function (Eq. (5)). In our simulations, the solvent particles are modeled through these three primary interaction potentials. On the other hand, for the colloidal particles, the conservative forces are excluded, and replaced by two additional interaction mechanisms: the hydrodynamic forces and an interparticle attraction. $\mathbf{F}^H$ represents a short-ranged lubrication force which depends on the drag term with $h_{ij}$ as the surface-surface distance between two colloidal particles. The pair drag term itself is defined as: $\mu_{ij} = 3\pi\eta_0 a_1 a_2/2a_{ij}$ where $a_1$ and $a_2$ are the radii of the interacting colloids. A short-ranged attractive potential is also used to simulate the colloidal particles suspended in a fluid[51]. The short-ranged attractive interactions are modeled through a Morse potential:

$$U_{\text{Morse}} = U_0(2e^{-\kappa h_{ij}} - e^{-2\kappa h_{ij}}), \tag{7}$$

where $\kappa^{-1}$ is the range of attraction. Both the lubrication and the attraction forces are calculated only between two colloidal particles.

After gelation under quiescent conditions, colloidal structures formed are imposed to Lees–Edwards boundary condition with different deformation rates[52,56]. The stress response of the material is then calculated using the Irving–Kirkwood formalism:

$$P = \frac{1}{V}\left\{ \sum_{i=1}^{N} m_i(\mathbf{v}_i - \mathbf{u}(\mathbf{r}_u)) \otimes (\mathbf{v}_i - \mathbf{u}(\mathbf{r}_u)) + \sum_{j>i}^{N}\sum_{i=1}^{N-1} \mathbf{r}_{ij} \otimes \mathbf{F}_{ij} \right\} \tag{8}$$

The first summation on the right-hand side of Eq. (8) measures the kinetic energy of individual particles and the second summation is the virial definition based on all individual pairwise interactions, $\mathbf{F}_{ij}$[57,58]. Here, We first decompose the total stress based on the particle type, i.e., solvent and colloid particles. For the solvent particles, the kinetic energy of the solvent phase is calculated over only the solvent particles, and for the virial term, $\mathbf{F}_{ij} = \mathbf{F}_{it}^C + \mathbf{F}_{ij}^D + \mathbf{F}_{ij}^R$; For the colloid phase, the

kinetic energy is again calculated over the colloidal particles and for the calculation of the virial term, in case of the colloid–colloid interactions, the calculated stress is calculated as, $\mathbf{F}_{ij} = \mathbf{F}_{ij}^D + \mathbf{F}_{ij}^R + \mathbf{F}_{ij}^M + \mathbf{F}_{ij}^H$, while for the case of the solvent-colloid interactions, the attractive force, $\mathbf{F}_{ij}^M$ and the lubrication force, $\mathbf{F}_{ij}^H$ are not calculated anymore. This designation enables an accurate calculation of the total stresses for each phase, as well as each interaction type. Namely, the colloid–colloid Morse interactions provide the attractive forces in the system, and all remaining interactions construct the acting hydrodynamic forces. It should be noted that the random force due to its nature sums to a zero total contribution and thus is excluded from this measurement, although calculated within the DPD formalism.

**Simulation parameters.** Prior to imposing shear flow, colloidal particles with a volume fraction of ($\phi = 0.2$) form space-spanning gel networks. The simulation box includes 518,400 solvent particles and 10,313 colloidal particles where simulation box size is 60 times the particle radius ($a = 1$) in all directions, the number density of ($\rho = 3$) is used for the solvent particles at a dimensionless temperature of ($k_B T = 0.1$). Densities are matched by setting the mass of a colloidal particle to ($m_C = 4/3\rho\pi a^3$), with the solvent particles of unit mass ($m_S = 1.0$). The strength of attraction potential is set to $U_0 = 6k_B T$ for the weak gel, and $20k_B T$ for the strong gel, with $\kappa = 33$ acting over a range of $0.1a$ to achieve a short-ranged weak attraction. Once colloidal gels are prepared, a constant shearing protocol is applied in a Mason range of $0.003 < \text{Mn} < 3.5$.

## Data availability

Additional data that support the findings of this study are available from the corresponding authors upon reasonable request. Source data are provided with this paper.

## Code availability

Simulations are performed using HOOMD-blue, the open-source molecular dynamics simulation toolkit, which is publicly available at the developers' website http://glotzerlab.engin.umich.edu/hoomd-blue/.

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

## Acknowledgements

We acknowledge funding from the American Chemical Society's Petroleum Research Fund (ACS PRF 60707). We wish to also thank Mr. Mehrshad Zandigohar for his invaluable technical help.

## Author contributions

M.N. and S.J. designed the computational framework, performed the calculations, analyzed the results, and wrote the manuscript.

## Competing interests

The authors declare no competing interests.
