## [Peer Review File · Nature Communications]

REVIEWER COMMENTS

Reviewer #1 (Remarks to the Author):

I have read this manuscript "Life and death of colloidal bonds control the rate-dependent rheology of gels" with much interest. The authors present numerical simulation results about the microstructural and micromechanical behaviour of attractive colloidal gels under shear deformation. They systematically study structure and mechanics upon varying what they call "Mason" number, ie the ratio between shear rate and attraction energy. The authors are able to cleverly decouple various contributions to the stresses, in particular bonding-attraction contributions are disentangled from hydrodynamic interaction contributions. This allows the authors to establish three different regimes, a compaction regime at low Mn number, a competition regime where hydro and attraction are both important and finally a fluidization regime at large Mn numbers. I think the results are overall important and deserve being published in a journal like Nature Communications, because they provide crucial new insights for the more microscopic understanding of a wide range of complex fluids under flow conditions. However, some changes are needed in order to make the paper suitable for Nature Commun., as listed below.

- The final message of the papers should be more focused and sharpened. The observations about the different regimes and interplay between attraction and hydrodynamics and bond-breakup are crucial and novel, but out of these phenomenology the authors should distillate a final message about the mechanism of yield-stress fluidization based on microscopic bond-breaking and hydrodynamic effects. For example fluidization appears to occur (?) when the red and blue curve become comparable on Fig.1(a). At a comparable value of Mn number, in Fig. 2b there appears to be a kink in the coordination number, when Z is about 5.0. Is there a relation between these two observations, and is this relevant for our understanding of fluidization in yield-stress systems?
- Regarding the low Mn compaction regime, I would expect to see Z grow a little bit, but this is not visible in Fig. 2b. Is this possibly due to how the bonding is defined in terms of cut-off distance for a bond to occur?
- When the authors introduce the Mn number some references where this number is derived/introduced are needed. Why is it called Mason number? A derivation based on dimensional analysis of what the authors call the Mn number can be found in Soft Matter 6, 2692-2698 (2010).
- The relation between bond lifetime and shear rate and/or bonding attraction should be compared with theoretical predictions (based on Kramers escape rate model extended to systems in shear flows) from Phys. Rev. E 87, 032310 (2013). In general, also the whole context of interplay between flow and particle interactions/dynamics, could be connected with contemporary theoretical research, in particular the crossover from interaction-dominated particle dynamics to flow-dominated has been derived and highlighted by approaches based on convection-diffusion equations, see Phys. Rev. E 80, 051404 (2009).
- Ref. [13] is largely the experimental verification of a theoretical model and predictions of Phys. Rev. Lett. 103, 208301 (2009), about the role of cluster structure on elasticity of colloidal gels.
- In Eq. 8, please define F^T . Is this the sum of the various force contributions (hydro, attractive bonding...)?
- In Eq. 4 I don't see how the conservative forces are connected to the interparticle potential Eq. 8... please clarify and provide details.

Reviewer #2 (Remarks to the Author):

In their manuscript, Nabizadeh and Jamali report on simulations performed to investigate the link between structural properties and the rheological response of colloidal gels. The structure of colloidal gels shows characteristic features on many length scales, from single particles and their interactions via a heterogeneous network structure on a mesoscopic length scale to the bulk

properties. In addition, because gels are non-equilibrium states, their structure and properties might depend on the preparation procedure. Hence to find and properly describe the *crucial* structural elements is very difficult. In particular, although the structure is accessible on all length scales by simulations and experiments, the link to bulk, e.g. rheological, properties is still not fully understood. This is despite the fact that gels are also frequently used in other areas of science and especially their mechanical, i.e. rheological, properties are technologically relevant. Thus, any advance in the understanding is very welcome and can have a very significant impact on the field as well as on a broad range of disciplines.

The present manuscript focuses on short-range attractive particle-particle interactions ('bonds') between colloidal particles and their relation to the rheological behaviour. It is suggested that the bond lifetime or mileage are correlated to the rheological response with long-lived and short-lived neighbors dominating the rheological response at low and large Mason numbers respectively, where the Mason number describes the balance between shear and interactions. The interplay between hydrodynamic stress and interparticle attraction is considered to be the key.

In the following I will discuss the main claims.

Instead of the bond age or life expectancy τ_B , the mileage of a colloidal bond, γ_B , is introduced as a new concept with different rheological parameters presented as a function of the mileage (Fig. 4). However, the mileage is trivially linked to the life expectancy through the shear rate $\dot{\gamma}$. Furthermore, the mileage is discussed in terms of "young bonds", "middle-aged bonds" and "old bonds" (toward end of page 5), i.e. words describing an age rather than a mileage. I thus do not understand why the concept of a mileage needs to be introduced and what can be gained by using this concept.

Moreover, a bond does not change with age or mileage. (In contrast to the structure of the gel, which might age due to its non-equilibrium state.) Thus, all bonds are identical and cannot be distinguished by their age. (Nevertheless, particles escape from their 'bonds' after different times but this is not due to different bonds that, e.g., have different ages but this is due to statistics as described in, e.g., ref. 40.)

In the manuscript, a central parameter is the Mason number that relates shear to interactions. It essentially is the ratio of the shear rate normalised by the (free) Brownian time to the strength of the interaction potential normalised by the thermal energy. However, only one strength of the potential $U_0 = 6 kT$ has been investigated. Thus, instead of the Mason number also the normalised shear rate could be used without losing anything. To show that the Mason number is a useful parameter, at least two strengths need to be studied.

It is claimed that based on the steady-state properties, the strain at the stress overshoot in a transient start-up test can be explained. However, as far as I understand, simulations of start-up tests are not presented and the predictions are not compared to corresponding simulation results. The prediction is compared to previous experimental results (Fig. 4d) where I could not find any information on the experiments or even a citation to the corresponding reference. Thus, it is not clear what has been compared. This is crucial as gels can show two overshoots (which one has been considered?) and the strain at the overshoot depends on the particle concentration, the strength of the interaction etc. (see, e.g., Figs. 9-11 in ref. 40 or Figs. 9-11 in J.Rheol. 55, 673).

In general, the methodology seems appropriate and the data appear sound. However, to transfer the results and concepts to different systems or other areas and to arrive at specific predictions, a more quantitative description is required. For some of the presented aspects, there are quantitative concepts available (and are cited). The opportunity to exploit them, unfortunately, has been missed.

Finally, some important information and explanations are not given in the manuscript, for example

- details of and reference to the experimental study, as mentioned above (page 8, Fig. 4d).
- definition of the diffusion time (page 2). Is this identical with τ_0 (page 4)?
- has it been checked that the steady state is reached after 500 diffusion times (page 2)?
- how are the hydrodynamic interactions determined (page 3, Fig. 1).

- color code is missing (Fig. 1d-g).
- decrease of the average contact number $\langle Z \rangle$ is described as non-monotonic (page 4) although Fig. 2b suggests otherwise.
- tic marks are missing in the inset to Fig. 2a.
- why can pre-existing bonds be neglected (page 4)?
- the exponential fits to the data should be shown (Fig. 3a).
- data in Fig 3a is shown as lin-lin (main figure) and log-log (inset) but for an exponential decay log-lin would be appropriate.
- τ_B is only implicitly defined in Fig. 3b.
- how is the data in Fig. 3c obtained? Are they just the inverse of the data in Fig. 3b?
- is the cited range (0.1) compatible with $\kappa = 33$ (page 9)?

General remarks:

We are very pleased with the interest our work has generated and the thoughtful reviews which provided a number of helpful suggestions for improvement and clarification. We were especially pleased to note that Reviewers A considers that: “...I think the results are overall important and deserve being published in a journal like Nature Communications, because they provide crucial new insights for the more microscopic understanding of a wide range of complex fluids under flow conditions”, and Reviewer B points out the significance of the topic discussed in our manuscript: “...any advance in the understanding is very welcome and can have a very significant impact on the field as well as on a broad range of disciplines”.

We have made numerous revisions to the manuscript, based on all comments and criticisms raised by the reviewers. The additional text is colored in red in the manuscript and is also included in the full response here. Our direct response to individual reviewer’s comments are also colored here in blue for ease of tracking, accompanied by changes made to the manuscript to facilitate the review process. We provide an extensive response to the feedback provided by both referees, and specifically made a number of additional changes and studies on a strongly-attractive system ($20 kT$ attraction strength compared to $6 kT$ attraction previously reported in the first version of the manuscript), a number of theoretical considerations, as well as results for flow-start up response of the colloidal gels. We have also tried harder to provide a clearer explanation of the new insights provided by our results as well as the novelty and merit of our approach, in a point by point manner. We believe that in the revised manuscript, and in our rebuttal letter, we address virtually all concerns and questions raised by the reviewers. We look forward to the reviewers’ assessment of this revised manuscript and the additional technical information.

Response to comments by Reviewer A:

I have read this manuscript "Life and death of colloidal bonds control the rate-dependent rheology of gels" with much interest. The authors present numerical simulations results about the microstructural and micromechanical behaviour of attractive colloidal gels under shear deformation. They systematically study structure and mechanics upon varying what they call "Mason" number, ie the ratio between shear rate and attraction energy. The authors are able to cleverly decouple various contributions to the stresses, in particular bonding-attraction contributions are disentangled from hydrodynamic interaction contributions. This allows the authors to establish three different regimes, a compaction regime at low Mn number, a competition regime where hydro and attraction are both important and finally a fluidization regime at large Mn numbers. I think the results are overall important and deserve being published in a journal like Nature Communications, because they provide crucial new insights for the more microscopic understanding of a wide range of complex fluids under flow conditions. However, some changes are needed in order to make the paper suitable for Nature Commun., as listed below.

We would like to thank the reviewer for thoughtful assessment of our manuscript. In the following we describe the changes made based upon suggestions/critisims raised by the reviewer.

Q- The final message of the papers should be more focused and sharpened. The observations about the different regimes and interplay between attraction and hydrodynamics and bond-breakup are crucial and novel, but out of these phenomenology the authors should distillate a final message about the mechanism of yield-stress fluidization based on microscopic bond-breaking and hydrodynamic effects. For example fluidization appears to occur (?) when the red and blue curve become comparable on Fig.1(a). At a comparable value of Mn number, in Fig. 2b there appears to be a kink in the coordination number, when Z is about 5.0. Is there a relation between these two observations, and is this relevant for our understanding of fluidization in yield-stress systems?

A- In a previous study, Jamali et al. PRL 2017, we had focused and discussed in detail the origin of the three regimes observed with respect to the strength of hydrodynamic and attractive forces. Nonetheless, that study was limited to the use of Mason number for such comparison and did not break down the contribution of different interactions to measure the actual values. In Fig. (1.a) the shear stress borne in the fluid and the colloidal phases are the consequence of fluidization, rather than its cause. Fluidization directly depends on the domintation of overall shear stress by the hydrodynamic forces. This means that for yielding to occur, the hydrodynamic interactions should become significant. However, as long as attractive forces are comparable in magnitude with the hydrodynamic forces, there will be secondary structure formation and transient kinematic heterogeneities within the flowing suspension. With regards to the coordination number changes, please see the response to the next question for a complete answer. Thus here, as suggested by the referee, we have added a number of clarifying sentences to sharpen these messages throughout the manuscript and in the conclusion section:

page 3: "The results in Fig. 1 strongly suggest that for the gel structure to yield, the hydrodynamic forces have to become comparable in magnitude with

the attractive forces; Nonetheless, the secondary structure formation, i.e. rejuvenation regime, prolongs until the hydrodynamic forces completely dominate the micromechanics of the system at which point the suspension becomes fully fluidized.”

page 7:“Our results show that the magnitude of the hydrodynamic force plays the central role in yielding of the structure. It should be noted that the Mason number alone provides a particle-level force scaling, and thus is not an accurate measure of the actual forces acting on the individual, clusters or networks of particles. As such, the screening of the hydrodynamic interactions for particles buried within the clusters, many-body effects and collective motion of colloidal aggregates will profoundly change the actual forces acting on different particles. Our study, leveraging the pairwise nature of interactions introduced, measures and decouples the overall acting forces on ensembles of particles.”

Q- Regarding the low Mn compaction regime, I would expect to see Z grow a little bit, but this is not visible in Fig. 2b. Is this possibly due to how the bonding is defined in terms of cut-off distance for a bond to occur?

A- The reviewer is absolutely correct in pointing out the expected growth of average coordination number in the shear compaction regime. To best show the compaction, one would need to alternatively show the scaled coordination number, normalized by the pseudo-equilibrium value of Z for a gel at the quiescent conditions. At quiescent conditions, and with the attractive interaction introduced here, the average contact number, $Z \approx 5.4$, and for the three smallest Mn values, shear compaction is indeed observed: the average contact number at quasi-steady state exceeds that of the quiescent conditions. We have made a figure below to show the scaled coordination number, plotting the average contact number at different Mn normalized by the average contact number of our gel at quiescent conditions. This trend is not dependent on the contact criteria, and thus changing the cut-off distance for bonding will not change the trends but will shift the curve vertically to different Z values. This is shown in figure where the average contact number is measured for a rather wide range of cut-off distances. Nonetheless it should be noted that the value of Z, and its growth over the shear compaction regime strictly depends on the volume fraction of particles in the system. To better explain this difference, one can consider the ultimate case of attractive glass, where using similar criterion for bonding, virtually no change in the value of Z would be observed. At lower volume fractions, the growth of Z in shear compaction regime becomes more visible. This is clearly visible by looking at the same graph for volume fraction of $\phi = 0.15$ reprinted from Jamali et al. Materials Today Advances, 2020. Accordingly, we have added some clarifying text to the manuscript to better explain this:

page 4:“In the shear compaction regime, the average value of the coordination number for the entire system grows slightly above its measure at the quiescent conditions, as indicated in the insert graph of Fig. 2b. However, the extent of this increase depends upon the volume fraction of particles, and is more visible for the lower fraction of colloids in the system.”

Q- When the authors introduce the Mn number some references where this number is derived/introduced are needed. Why is it called Mason number? A derivation based on dimensional analysis of what the authors call the Mn number can be found in Soft Matter 6, 2692-2698 (2010).

A- Mason number was first introduced in the study of the electrorheological fluids to measure the ratio of viscous forces that tend to erupt the microstructures that are formed due to the electric field interparticle interactions.¹ Mason and co-workers showed that the intimate rheological response of suspensions -where colloidal particles form microstructures due to interparticle interactions- could be explained by generalization of Peclet number to $Mn = Pe/\beta$, where $Pe = 6\pi\eta_0 a^3 \dot{\gamma}/k_B T$ and β is a dimensionless group that characterizes the importance of the bond making interparticle interactions². In the context of attractive colloidal gels, Mn is adapted to measure shear to attraction potentials where the latter is modeled through Morse potential, here the attraction well depth is comparable to the system temperature hence we think Mn is a suitable dimensionless group for our work. We would like to thank the referee for pointing out the use of same concept in definition of a Breakage number by Xie et al.. The aforementioned study, as well as a number of other works by Koumakis, Petekidis, Ahn, Wagner and others have used the same concept and have referred to the dimensionless group as Peclet of depletion or inverse of Bingham number. We have modified the text in the manuscript early on and in the introduction section to make this clarification.

page 2:“There is a growing body of literature now suggesting that it is the competition between the interparticle attractive forces and the hydrodynamic shearing forces that determines the state of the particulate microstructure and hence the mechanical response of a colloidal gel³⁻¹¹. This competition can be understood using a dimensionless group referred to as the Mason number, ($Mn = 6\pi\eta_0 \dot{\gamma} a^3 / U_0$), in which the η_0 , $\dot{\gamma}$, a , and U_0 are solvent viscosity, applied deformation rate, particle radius, and the strength of attractive potential between the particles. At small $Mn < 0.01$, the shearing flows do not result in fluidization of the particulate structure and in contrast lead to shear compaction of the structure. At intermediate $0.01 < Mn < 1$ the consistent breakage and reformation of particle-particle bonds result in structural and kinematic heterogeneities to emerge, and ultimately at large $Mn > 1$, the shearing forces effectively break the particle bonds resulting in a fluid-like flow similar to a suspension of non-interacting colloids⁶. Mason number, first introduced for electrorheological fluids^{1,2} is used throughout this study. However, a Buckingham theorem dimensional analysis of similar force comparison was proposed by Xie et al.¹², suggesting a Breakage number, Br . Additionally, in a series of previous studies Peclet of depletion, Pe_{dep} , was introduced by Koumakis and coworkers^{13,14}

and later used by others^{7,8,11}. While these dimensionless groups are all similar in essence, we use Mason number as the dimensionless group of choice as it provides a foundation to compare the magnitude of two competing forces in the system.”

Q- The relation between bond lifetime and shear rate and/or bonding attraction should be compared with theoretical predictions (based on Kramers escape rate model extended to systems in shear flows) from Phys. Rev. E 87, 032310 (2013). In general, also the whole context of interplay between flow and particle interactions/dynamics, could be connected with contemporary theoretical research, in particular the crossover from interaction-dominated particle dynamics to flow-dominated has been derived and highlighted by approaches based on convection-diffusion equations, see Phys. Rev. E 80, 051404 (2009).

This is a good point, and we have now taken note of that in the manuscript. According to Kramer’s escape rate model, in an inter-particle interaction dominated regime, the break-up rate of bonds changes as a power law function of the shear rate, while an exponential form is expected for a shear dominated regime. Nonetheless, theoretically one can make such calculations based on a pair of particles and without consideration of many body effects, or for a given size of particle cluster based on a given fractal dimension. In a real colloidal gel, a distribution of different cluster sizes, bonds and particle populations exist, resulting in non-trivial hydrodynamics that cannot be simply explained through Stokes’ drag type of relation. Additionally, the theory refers to the rate of bond breakage, and the entire process, as opposed to a steady state window studied in our work. Nonetheless, our results, based on total rates of bond breakage and formation qualitatively agree with the theoretical predictions, specially in the transition regime from interaction to shear dominated regime. This is now more clearly observed in log-log plots of bond breakage and formation in Fig. 3c, which includes two different ranges of attraction. The rupture rate of bonds for both the strong ($D_0 = 20kT$) and weak ($D_0 = 6kT$) gels are dependent to shear rate through power-law relations. According to Kramer model, the power-law dependent rupture rate is expected only for the interparticle interaction dominated regime, i.e. low Mn . For the shear dominated regimes, Kramer model predicts an exponential growth of the rupture rate, in contrast with our results of the rather collision-controlled nature of the mixture at highest deformation rates. We see in Fig. 1.c that the hydrodynamic stresses become comparable to the attractive ones around $Mn = 0.2$, this point also coincides with the point where rupture rates of the two gels become comparable. (see Fig. 3.c). Interestingly, at larger Mn , for both attraction levels, the rate of rupture seems to be controlled only by the shear rate, still through a power-law dependence, with a similar slope for both the weak and the strong gels. The similarity between the rupture rate/shear rate relation suggests the similarity of the kinetics of our gels under highly hydrodynamic dominated conditions. This is also consistent with Fig. 1a,bc because at high Mn , we’ve shown the Newtonian-like behavior of gel where shear rate is solely responsible for the resulting shear stress through the hydrodynamic interactions. For higher Mn , the rupture rate of bonds of the strong gel ($D_0 = 20kT$), become more frequent than that of the weak gel ($D_0 = 6kT$). This is expected, as a higher shear rate is required for the stronger gel, to achieve the same Mn , compared to the weak gel. We have now added a clear citation of the original work as well as the discussion in the revised manuscript as follows:

Page 5-“...In the shear compaction regime each particle loses $\sim 0.002Z$ of its

contacts during each diffusion time, in stark contrast to more than $0.05Z$ in the fluidization regime (Fig.3.c). Nonetheless, this average rate representation does not mean that long-lived bonds do not exist at high Mason numbers. Furthermore the results for different attractions strengths show that while the rate of formed/broken bonds change for different values of attraction, the behavior shows a similar trend when plotted against the Mason number. The results in Fig. 3c, and alternatively the bond life times, can be also compared to the theoretical framework based on Kramer’s escape rate model¹⁵. Based on this theory, rate of aggregate breakage shows a power-law dependence on the applied deformation rate, transitioning to an exponential function in a shear-controlled regime. While such calculations do not include the complex many body dynamics and their resulting hydrodynamics on the bond breakage and formation for the interaction dominated regime. Our results, based on total rates of bond breakage and formation in the quasi steady state regime qualitatively agree with the theoretical predictions. In Fig. 3c, the rupture/formation rate of bonds for both the strong ($D_0 = 20kT$) and weak ($D_0 = 6kT$) gels depend on the applied Mason number through a power-law function. The contribution of hydrodynamic stresses become comparable to the ones from attractive forces at $\sim Mn = 0.2$ (Fig. 1c), coinciding with the point where rupture rates of the two gels become comparable (Fig. 3.c). For higher Mn, the rupture rate of bonds of the strong gel ($D_0 = 20kT$), become more frequent than that of the weak gel ($D_0 = 6kT$). This is expected, as a higher shear rate is required for the stronger gel, to achieve the same Mn , compared to the weak gel...”

Q- Ref. [13] is largely the experimental verification of a theoretical model and predictions of Phys. Rev. Lett. 103, 208301 (2009), about the role of cluster structure on elasticity of colloidal gels.

We would like to thank the referee for bringing our attention to the original theoretical work, we have added a sentence to the manuscript and the proper citation:

Page 1 - “Originally, theoretical and computational investigations¹⁶ and more recently an experimental work⁴ have shown that these locally glassy structures are the origin of elasticity in colloidal gel networks.”

Q-In Eq. 8, please define F^T . Is this the sum of the various force contributions (hydro, attractive bonding...)?

A - F_{ij}^T represents the total force between two interacting particles; When decomposing the total stress to the hydrodynamic and attractive components, the only force that is considered in the attractive interaction being the Morse potential, all the other interactions are considered as hydrodynamics, i.e. dissipative force, random force, short-range lubrication force. It’s worth mentioning that we observe in our results that the resulting stress of the random force is negligible for all regimes of Mason, and the hydrodynamic stresses mainly originate from the dissipative and lubrication forces. Nonetheless, we have changed the notion to a general pairwise one to avoid any confusion:

“The stress response of the material is then calculated using the Irving-Kirkwood formalism:

$$P = \frac{1}{V} \{ \sum_{i=1}^N m_i (v_i - u(r_u)) \otimes (v_i - u(r_u)) + \sum_{j>i}^N \sum_{i=1}^{N-1} r_{ij} \otimes F_{ij} \} \quad (1)$$

The first summation on the right hand side of Eq. 1 measures the kinetic energy of individual particles and the second summation is the virial definition based on all individual pairwise interactions, F_{ij} ^{17,18}.”

Q- In Eq. 4 I don't see how the conservative forces are connected to the interparticle potential Eq. 8... please clarify and provide details.

A- Here, $F_{ij} = F_{ij}^C + F_{ij}^R + F_{ij}^D + F_{ij}^H + F_{ij}^M$. Here, the colloidal particles do not interact via the conservative force, hence there is no contribution to the stress of the colloidal phase that could come from the conservative forces. For the solvent-solvent and solvent-colloid interactions, the conservative force is solved and considered in the stress calculation. We also observe that the conservative force always leads to a small portion of the total stress, i.e. $\max(\sigma_{xy}^C/\sigma_{xy} < 0.01)$. We re-wrote the details of interaction potentials to make it more clear:

“We write the equation of motion for the DPD formalism as:

$$m_i \frac{d\mathbf{v}_i}{dt} = \sum_{i, i \neq j}^{N_p} (\mathbf{F}_{ij}^C + \mathbf{F}_{ij}^D + \mathbf{F}_{ij}^R + \mathbf{F}_{ij}^H + \mathbf{F}_{ij}^M) \quad (2)$$

...

$$\omega_{ij} = (1 - r_{ij}/r_c) \quad (3)$$

$$\mathbf{F}_{ij}^H = \mu_{ij}^H (\mathbf{v}_{ij} \cdot \mathbf{e}_{ij}) \mathbf{e}_{ij}, \quad (4)$$

... In our simulations, the solvent particles are modeled through these three primary interaction potentials. On the other hand, for the colloidal particles, the conservative forces are excluded, and replaced by two additional interaction mechanisms: the hydrodynamic forces and an interparticle attraction. \mathbf{F}^H represents a short-ranged lubrication force which depends on the drag term with h_{ij} as the surface-surface distance between two colloidal particles. The pair drag term itself is defined as: $\mu_{ij} = 3\pi\eta_0 a_1 a_2 / 2h_{ij}$ where a_1 and a_2 are the radii of the interacting colloids... Both the lubrication and the attraction forces are calculated only between two colloidal particles.”

Response to comments by Reviewer B:

In their manuscript, Nabizadeh and Jamali report on simulations performed to investigate the link between structural properties and the rheological response of colloidal gels. The structure of colloidal gels shows characteristic features on many length scales, from single particles and their interactions via a heterogeneous network structure on a mesoscopic length scale to the bulk properties. In addition, because gels are non-equilibrium states, their structure and properties might depend on the preparation procedure. Hence to find and properly describe the *crucial* structural elements is very difficult. In particular, although the structure is accessible on all length scales by simulations and experiments, the link to bulk, e.g. rheological, properties is still not fully understood. This is despite the fact that gels are also frequently used in other areas of science and especially their mechanical, i.e. rheological, properties are technologically relevant. Thus, any advance in the understanding is very welcome and can have a very significant impact on the field as well as on a broad range of disciplines.

The present manuscript focuses on short-range attractive particle-particle interactions ('bonds') between colloidal particles and their relation to the rheological behaviour. It is suggested that the bond lifetime or mileage are correlated to the rheological response with long-lived and short-lived neighbors dominating the rheological response at low and large Mason numbers respectively, where the Mason number describes the balance between shear and interactions. The interplay between hydrodynamic stress and interparticle attraction is considered to be the key.

In the following I will discuss the main claims.

Q- Instead of the bond age or life expectancy τ_{B} , the mileage of a colloidal bond, γ_{B} , is introduced as a new concept with different rheological parameters presented as a function of the mileage (Fig. 4). However, the mileage is trivially linked to the life expectancy through the shear rate. Furthermore, the mileage is discussed in terms of "young bonds", "middle-aged bonds" and "old bonds" (toward end of page 5), i.e. words describing an age rather than a mileage. I thus do not understand why the concept of a mileage needs to be introduced and what can be gained by using this concept. Moreover, a bond does not change with age or mileage. (In contrast to the structure of the gel, which might age due to its non-equilibrium state.) Thus, all bonds are identical and cannot be distinguished by their age.

A- We believe that there has been a misunderstanding in our arguments made in the manuscript and thus here we will try to clarify those for the reviewer. First of all, we would like to thank the referee for pointing out the inconsistencies in the choice of terminology describing the life expectancy and bond mileage. We have now modified those terms to "low mileage", "medium mileage" and "high mileage" bonds to emphasize on the notion of bond mileage rather than lifetime. Nonetheless, the bond mileage concept is a very important point that we would like to better describe here. It is widely accepted that the rheology of these gels are not only rate dependent, but also time dependent, as they exhibit a wide range of thixotropic behavior under different flow regimes. This is manifested very clearly in large rheological hysteresis observed in colloidal gels (experiments by Divoux et al. PRL 2013, and computationally shown in Jamali et al. PRL 2019). In a recent work we also showed that for these short-range attractive system, the final mechanical response of the suspension, when

brought to rest from flowing condition, can be described based upon the total strain that the particles have experienced (Jamali et al. *Materials Today Advances*, 2020). Figures reprinted from this previous work below show how for a wide range of attraction strengths and ranges, using a total value of accumulated strain unifies the coordination number measure, number density fluctuations as well as the yield/residual stresses in the system. In both figures, the y-axis are normalized by the same measure for a quiescent gel, and the x-axis represents the total normalized value of accumulated strain [by the range of attraction] that the colloids have experience before being brought to rest. For further details please see the publication cited. Thus it is crucial to provide a measure of the bonding and its resulting mechanics,

that is dimensionless, and reflective of the history of flow in addition to the rate of imposed flow. The life-expectancy in our study identifies the average life time of a colloidal bond under an applied deformation. That is the time, that all bonds on average, would stay alive under an applied deformation. However, the bond mileage calculated in our work is in fact from the actual number of strains that different bonds have endured under shear, and is not simply the product of average life expectancy and the rate of deformation. In other words, that product yields an average bond mileage for the system, but does not present a correlation to which bonds (low mileage or high mileage) are the main source of stress in the system. On the other hand, the life expectancy measure, by definition has units of time, which again should be compared to the diffusion time or the total time of flow to deduce any conclusions. Hence, the bond mileage conceptualizes both the duration of flow, as well as the strength of flow, in a unified fashion to provide information on where largest stresses are borne. Additionally, bond mileage of a bond represents the number of shear strains under which the bond has survived, and we only consider bonds that are formed under shear and at quasi-steady state. The referee is absolutely correct in the fact that having short-range interactions with identical densities, one can not distinguish the individual bonds based on their lifetime; however, that in fact is the very advantage of having access to all bonds at all times and interrogating which bonds at each instance borne largest stresses in the system. Our previous work, using fabric tensor to look at the orientation of individual bonds in a system (Jamali et al., PRL 2017), we showed that when presented in the context of strain, different components of the particulate network fabric show a universal behavior regardless of the applied deformation rate and the flow regime. The figure below shows the shear component as well as the deviatoric components of the fabric tensor, for an attractive colloidal system under flow at different magnitudes of shear rate applied, showing a universal behavior when presented against strain. Therefore, our bond mileage concept is the only tool

that connects the population of bonds with similar flow histories to the macroscopic stresses that are measured, independent of the shear rate applied. For instance, we observe that, statistically, a bond needs to survive $\gamma \approx 1$ strain, in order to positively contribute to the attractive stresses, which is the average amount of strain it takes for a newly formed bond -in an arbitrary angle- to orient in the shearing direction. Hence, we completely agree with your point on the indistinguishable nature of individual bonds, and we emphasize on how bond populations of the same lifetime can be distinguished by their microscopically measured stresses. Nonetheless we have also added the following discussions to our manuscript to clarify this distinction, and the need for using bond mileage:

Page 5-“...For systems with $\tau_B > 400\tau_0$, attractive forces are the primary source of macroscopic stress and by dominating the hydrodynamic forces effectively increase the bond life expectancy. On the contrary, short-lived neighbors with $\tau_B < 4\tau_0$ are controlled by the hydrodynamic forces. Bonds with the intermediate life expectancy of $6\tau_0 < \tau_B < 400\tau_0$ are however controlled through an interplay between the hydrodynamics and attractive forces. Now and with the dynamical bond lifetime information as well as the macroscopic stresses at hand, we can identify the particle bonds with respect to their contribution to the total macroscopic stresses in the material under flowing condition. The rheology of colloidal gels are not only rate dependent, but also time dependent, as they exhibit a wide range of thixotropic behavior under different flow regimes. For instance, the final mechanical response of the suspension, when brought to rest from flowing condition, can be described based upon the total strain that the particles have experienced⁵. Additionally, the fabric of the particulate network is found to exhibit a universal behavior with respect to its shear and deviatoric components, when presented against the shear strain instead of shear rate or Mason number⁶. While the life-expectancy identifies the average life time of a colloidal bond under an applied deformation, a strain measure of the populations that have experienced the same shear history would be beneficial. Accordingly, we define the total accumulated strain observed over the lifetime of each bond, as its bond mileage, γ_B . Fig. 4.a-b show the contributions of attractive and hydrodynamic forces to the total stress of the colloidal phase against the bond mileage at different Mn values, respectively. It should be noted that the bond mileages presented here are sampled over the entire range of quasi steady-state flow. To do this, all populations of bond life times and their corresponding stresses are

tracked and data are collected over the entire flow protocol...”

Q- In the manuscript, a central parameter is the Mason number that relates shear to interactions. It essentially is the ratio of the shear rate normalised by the (free) Brownian time to the strength of the interaction potential normalised by the thermal energy. However, only one strength of the potential $U_0 = 6kT$ has been investigated. Thus, instead of the Mason number also the normalised shear rate could be used without losing anything. To show that the Mason number is a useful parameter, at least two strengths need to be studied.

The referee is absolutely correct and we appreciate the suggestion. In order to justify the use of Mason number we performed a series of simulations with a range of attractions changing from $U_0 = 3kT$ to $U_0 = 30kT$, and now included the results for $U_0 = 20kT$ throughout the manuscript, as one of the most commonly studied (computationally) attraction strengths. Our results clearly show, specially in Figure 1, that the contributions to stress follow a very similar trend when presented in terms of Mason number. As such, we also added the average and normalized coordination numbers for the higher attraction strength of $U_0 = 20kT$ in figure 2 and 3. For ease of tracking we have also reproduced the same graphs here for your consideration, as well as changes to the manuscript accordingly:

Page 3-“...Results in Fig. 1 a-c also clearly show that when presented against Mason number, the total shear stress as well as individual contributions from the colloidal phase (both attractive and hydrodynamic forces) follow virtually identical trends for different strengths of attraction. While the yield stress of the colloidal network increases for higher attraction strengths, the source of stresses and their relative contribution remain unchanged, further suggesting that the Mason number appropriately classifies the rheological behavior of these materials...”

Page 4-“...In the first regime, $\langle Z \rangle$ remains virtually intact, as the dominating attractive forces effectively keep particles together and retain the structural integrity of particulate clusters. The dynamic and consistent breakage and formation of particle bonds results in a non-monotonic decrease of $\langle Z \rangle$ in the

intermediate Mn regime, followed by a terminal regime where $\langle Z \rangle$ can essentially be predicted using the collisional dynamics of particle population at each volume fraction. Furthermore, the comparison between the average number of bonds per particle under flow shows that while there are quantitative differences in the values measured for weak and strong attractions of $U_0 = 6kT$ and $U_0 = 20kT$, the trends when presented against the Mason number follow a universal behavior, as previously shown and discussed in⁷....”

Page 5-“...In the shear compaction regime each particle loses $\sim 0.002Z$ of its contacts during each diffusion time, in stark contrast to more than $0.05Z$ in the fluidization regime (Fig.3.c). Nonetheless, this average rate representation does not mean that long-lived bonds do not exist at high Mason numbers. Furthermore the results for different attractions strengths show that while the rate of formed/broken bonds change for different values of attraction, the behavior shows a similar trend when plotted against the Mason number ..”

Q- It is claimed that based on the steady-state properties, the strain at the stress overshoot in a transient start-up test can be explained. However, as far as I understand, simulations of start-up tests are not presented and the predictions are not compared to corresponding simulation results. The prediction is compared to previous experimental results

(Fig. 4d) where I could not find any information on the experiments or even a citation to the corresponding reference. Thus, it is not clear what has been compared. This is crucial as gels can show two overshoots (which one has been considered?) and the strain at the overshoot depends on the particle concentration, the strength of the interaction etc. (see, e.g., Figs. 9-11 in ref. 40 or Figs. 9-11 in J.Rheol. 55, 673).

This is also correct. Previously and since the focal point of our manuscript has been on the quasi steady state regime, we had not included the start-up results, but as those are clearly required to justify our results in Fig.4, we now have included a graph with the results where the strain at the stress overshoot is measured. Obviously these results are not unique, and previously ourselves^{6,7} as well as others^{11,13,14,19} have reported different mechanisms of yielding including a single step and two step yielding behavior in colloidal gels. While the two-step yielding and emergence of two different overshoots is not uncommon, here we have only focused on the first stress overshoot. The strain at which this overshoot is measured is reported to have different trends both in experiments and computation, but here for the sake of consistency we are measuring these strains as opposed to referring to other measurements. This ensures that the same system is being discussed. However, we have included the figure in the manuscript, and added some context as well as proper citation to experimental and computational studies of the same phenomenon. For ease of tracking we have also reproduced the same graphs here for your consideration, as well as changes to the manuscript accordingly:

Page 7-“...Most importantly, our results clearly indicate that these critical strains at which stress overshoot is observed, γ_C , also show the same trend and scaling as the maximum in the stress-bond mileage graphs. This is significant, since the bond mileages are all calculated in the quasi-steady state and far from this initial departure from the linear viscoelastic response, and yet reflect on the microstructural and dynamical sources of the stress overshoot. The critical strains are directly calculated from the initial flow simulations, where the stress response at the inception of shear flow is monitored. Previous computational and experimental efforts^{6,7,11,13,14,19} have reported in detail, different mechanisms of yielding including a single and two step yielding behavior in colloidal gels, also referred to the static and the dynamic yielding. While the two-step yielding and emergence of two different overshoots are not uncommon, here we have only focused on the first stress overshoot. The strain at which this overshoot is measured is reported to have different trends both in experiments and computation, but here

for the sake of consistency we are measuring these strains from our flow start-up simulations...”

Q- In general, the methodology seems appropriate and the data appear sound. However, to transfer the results and concepts to different systems or other areas and to arrive at specific predictions, a more quantitative description is required. For some of the presented aspects, there are quantitative concepts available (and are cited). The opportunity to exploit them, unfortunately, has been missed.

We believe that the new additional studies, including the new results on different attraction strengths, the flow start up measurements as well as the discussions, also the addition of a theoretical framework makes the manuscript more comprehensive and relevant for similar systems of interest.

Q- Finally, some important information and explanations are not given in the manuscript, for example:

- details of and reference to the experimental study, as mentioned above (page 8, Fig. 4d).

Please refer to our response to the previous question regarding the stress overshoot measurements. We have also updated the manuscript with proper citations.

- Q- definition of the diffusion time (page 2). Is this identical with τ_0 (page 4)?

Yes, diffusion time, τ_0 is the same value we mention in page 4. Diffusion time is defined the time it takes for a particle to move it’s own diameter when suspended in a fluid. We calculate this value in our simulation through the mean-squared displacement parameter. For this purpose, we simulate a single colloid suspended in the fluid, and calculate $MSD(t) = \langle |\mathbf{r}(t + \tau) - \mathbf{r}(t)|^2 \rangle$ over time (See Fig.). The diffusion coefficient, D , of the freely diffusing colloid is then calculated. The diffusion time is then calculated via $\tau_0 = \frac{a^2}{2D}$, where a is the radius of the colloid. We have provided the additional figure here for referee’s consideration on the MSD evolution of a bear particle.

- Q- has it been checked that the steady state is reached after 500 diffusion times (page2)?

A- Yes, here we check the steady state through both the contact distribution of the gel and the mean squared displacement of the colloidal phase. Of course, there will be

always a very slight change in particle contact numbers as the lively nature of these contacts, specifically at weaker attractions, result in constant breakage and formation of particle bonds. However both the MSD graph as well as the average value and distribution of contact numbers indicate a stable network formation at these times. Figures of both quantities measured over time are presented here for the referee.

- Q- how are the hydrodynamic interactions determined (page 3, Fig. 1).

A- Thank you for raising this question. We have added a explicit/detailed part to the "Methods" section, as follows:

Methods section: "...Here, We first decompose the total stress based on the particle type, i.e. solvent and colloid particles. For the solvent particles, the kinetic energy of the solvent phase is calculated over only the solvent particles, and for the virial term, $F_{ij} = F_{ij}^C + F_{ij}^D + F_{ij}^R$; For the colloid phase, the kinetic energy is again calculated over the colloidal particles and for the calculation of the virial term, in case of the colloid-colloid interactions, the calculated stress is calculated as, $F_{ij} = F_{ij}^D + F_{ij}^R + F_{ij}^M + F_{ij}^H$, while for the case of the solvent-colloid interactions, the attractive force, F_{ij}^M and the lubrication force, F_{ij}^H are not calculated anymore. This designation enables an accurate calculation of the total stresses for each phase, as well as each interaction type. Namely, the colloid-colloid Morse interactions provide the attractive forces in the system, and all remaining interactions construct the acting hydrodynamic forces. It should be noted that the random force due to its nature sums to a zero total contribution and thus is excluded from this measurement, although calculated within the DPD formalism..."

- Q- color code is missing (Fig. 1d-g).

A- Thanks for bringing this to our attention, the color code is added to Fig. 1d-g.

- Q- decrease of the average contact number $\langle Z \rangle$ is described as non-monotonic (page 4) although Fig. 2b suggests otherwise.

A- This was a bad choice of wording and we have changed that in the revised manuscript. We also have made an insert graph with the normalized value of the average coordination number against Mason number, to clarify the shear compaction regime.

page 4: "...In the shear compaction regime, the average value of the coordination number for the entire system grows slightly above its measure at the quiescent conditions, as indicated in the insert graph of Fig. 2b. However, the extent of this increase depends upon the volume fraction of particles, and is more visible for the lower fraction of colloids in the system..."

- Q- tic marks are missing in the inset to Fig. 2a.

A- We have fixed the issue and plotted the Fig. here for ease of tracking.

- Q- why can pre-existing bonds be neglected (page 4)?

A- The goal of our study is to study the rheology of gel under shear rather than the shear start-up. If we include the pre-existing bonds, the lifetime of bonds that have already existed before applying the shear, would create inconsistency in the population of bonds under study. This is particularly important as we are trying to understand the notion of rheology-structure coupling under flow, without being biased on the pre-existing bonds. Thus any life time calculation prior to the quasi steady state must be neglected from the calculation.

- Q- the exponential fits to the data should be shown (Fig. 3a).

A- We have made the changes suggested by the referee and added the exponential fits with dashed lines in the figure. The figure is also replotted here for ease of tracking and referee's consideration

- Q- data in Fig 3a is shown as lin-lin (main figure) and log-log (inset) but for an

exponential decay log-lin would be appropriate.

A- We have made the insert graph a log-lin graph per referee's suggestion, and have replotted the figure here for referee's consideration

- Q- τ_B is only implicitly defined in Fig. 3b.

A- We have added the following description to the manuscript with a clear definition of the life expectancy and how it is defined:

Page 5 - “The lifetime probability shows an exponentially decaying behavior for all Mn numbers. Here, we fit $P(\tau) = C \exp(-\tau/\tau_B)$, where C is a conversion factor constant, τ is bond lifetime and we name τ_B as the life expectancy.”

- Q- how is the data in Fig. 3c obtained? Are they just the inverse of the data in Fig. 3b?

A- No, the data in Fig. 3b is calculated from the exponential decay of the life time distribution functions in 3.a. In contrast, the at each Mn, once steady state is reached, we count the total number of bonds that are ruptured/formed per diffusion time, and normalize that by the total number of particles to calculate the rates in Fig. 3c. We have made a clarifying comment in the text as follows:

Page 5 - “...We define this characteristic timescale as the bond life expectancy. The bond life expectancy against the applied Mn is presented in Fig. 3.b, showing a power law dependence on the imposed deformation rate. The decrease in the life expectancy suggests a fast breakage/formation dynamic for the colloidal bonds, and does not necessarily imply a lower number of bonds. Also to be noted is the fact that these bond dynamics are measured at the quasi-steady state and far from yielding dynamics, where the rate of bond formations and bond breakages take different values as

suggested by¹⁹. The consistent and steady bond numbers can be verified in Fig.3.c, showing exactly the same rate of broken and formed bonds at different Mn values. Note that the results in Fig. 3b are measured from the characteristic relaxation time of life time distribution probabilities in Fig. 3a, while the rates of ruptured and formed bonds are measured directly from the simulations and monitoring the number of bonds per diffusion time...”

- Q-is the cited range (0.1) compatible with $\kappa = 33$ (page 9)?

The cited range is selected based on two reasons: First, the first peak of the radial distribution function, $g(r)$, of the final state gel appears around $h_{ij} \approx 0.1$ (see Fig. . a). Secondly, the Morse potential U_{Morse} equates the particle’s thermal energy at $h_{ij} \approx 0.074$, i.e. $U_0(e^{-2\kappa h_{ij}} - 2e^{-\kappa h_{ij}}) = k_B T$. (See Fig. . b). We choose the separation distance of $h_{ij} = 0.1$ as our contact criteria, rather than $h_{ij} = 0.074$ because iso-statistically, our microstructure analysis is not sensitive to the criteria in that range, i.e. the maximum change in the average contact number in all simulations is less than 5% (the highest Mn). On the other hand, in the context of lifetime, -regarding the oscillating nature of bonds- a slightly relaxed contact criteria helps us capture a more accurate measurement on the bond lifetime, where we allow bonds a slight threshold above their actual contact range before considering them broken. We also made a plot here showing the sensitivity of our results (average Z under each shear rate as a representative measure), based on different cut-off distances for bonding as evidence to unbiased nature of results presented in the manuscript.

References

- [1] Alice P Gast and Charles F Zukoski. Electrorheological fluids as colloidal suspensions. *Advances in Colloid and Interface Science*, 30:153–202, 1989.
- [2] IYZ Zia, RG Cox, and SG Mason. Ordered aggregates of particles in shear flow. *Proceedings of the Royal Society of London. Series A. Mathematical and Physical Sciences*, 300(1463):421–441, 1967.
- [3] Hajime Tanaka and Takeaki Araki. Simulation method of colloidal suspensions with hydrodynamic interactions: Fluid particle dynamics. *Physical review letters*, 85(6):1338, 2000.

- [4] Kathryn A Whitaker, Zsigmond Varga, Lilian C Hsiao, Michael J Solomon, James W Swan, and Eric M Furst. Colloidal gel elasticity arises from the packing of locally glassy clusters. *Nature communications*, 10(1):1–8, 2019.
- [5] Safa Jamali, Robert C Armstrong, and Gareth H McKinley. Time-rate-transformation framework for targeted assembly of short-range attractive colloidal suspensions. *Materials Today Advances*, 5:100026, 2020.
- [6] Safa Jamali, Gareth H McKinley, and Robert C Armstrong. Microstructural rearrangements and their rheological implications in a model thixotropic elastoviscoplastic fluid. *Physical review letters*, 118(4):048003, 2017.
- [7] Arman Boromand, Safa Jamali, and João M Maia. Structural fingerprints of yielding mechanisms in attractive colloidal gels. *Soft matter*, 13(2):458–473, 2017.
- [8] Esmaeel Moghimi, Alan R Jacob, and George Petekidis. Residual stresses in colloidal gels. *Soft Matter*, 13(43):7824–7833, 2017.
- [9] Zsigmond Varga, Vincent Grenard, Stefano Pecorario, Nicolas Taberlet, Vincent Dolique, Sébastien Manneville, Thibaut Divoux, Gareth H McKinley, and James W Swan. Hydrodynamics control shear-induced pattern formation in attractive suspensions. *Proceedings of the National Academy of Sciences*, 116(25):12193–12198, 2019.
- [10] Lilian C Hsiao, Richmond S Newman, Sharon C Glotzer, and Michael J Solomon. Role of isostaticity and load-bearing microstructure in the elasticity of yielded colloidal gels. *Proceedings of the National Academy of Sciences*, 109(40):16029–16034, 2012.
- [11] Esmaeel Moghimi and George Petekidis. Mechanisms of two-step yielding in attractive colloidal glasses. *Journal of Rheology*, 64(5):1209–1225, 2020.
- [12] Delong Xie, Hua Wu, Alessio Zaccone, Leonie Braun, Huanqin Chen, and Massimo Morbidelli. Criticality for shear-induced gelation of charge-stabilized colloids. *Soft Matter*, 6(12):2692–2698, 2010.
- [13] N Koumakis and G Petekidis. Two step yielding in attractive colloids: transition from gels to attractive glasses. *Soft Matter*, 7(6):2456–2470, 2011.
- [14] Nick Koumakis, Esmaeel Moghimi, Rut Besseling, Wilson CK Poon, John F Brady, and George Petekidis. Tuning colloidal gels by shear. *Soft Matter*, 11(23):4640–4648, 2015.
- [15] Alessio Zaccone, Hua Wu, Daniele Gentili, and Massimo Morbidelli. Theory of activated-rate processes under shear with application to shear-induced aggregation of colloids. *Physical Review E*, 80(5):051404, 2009.
- [16] Alessio Zaccone, Hua Wu, and Emanuela Del Gado. Elasticity of arrested short-ranged attractive colloids: Homogeneous and heterogeneous glasses. *Physical review letters*, 103(20):208301, 2009.

- [17] Aidan P Thompson, Steven J Plimpton, and William Mattson. General formulation of pressure and stress tensor for arbitrary many-body interaction potentials under periodic boundary conditions. *The Journal of chemical physics*, 131(15):154107, 2009.
- [18] DH Tsai. The virial theorem and stress calculation in molecular dynamics. *The Journal of Chemical Physics*, 70(3):1375–1382, 1979.
- [19] Jader Colombo and Emanuela Del Gado. Stress localization, stiffening, and yielding in a model colloidal gel. *Journal of rheology*, 58(5):1089–1116, 2014.

REVIEWER COMMENTS

Reviewer #1 (Remarks to the Author):

I am satisfied with the response of the authors to my comments and also found their response to the comments of the other referee highly satisfactory. I believe this is an important paper that will be of great interest to a broad audience, and am happy to recommend it for publication in Nature Communications in its present form.

Reviewer #2 (Remarks to the Author):

The manuscript has been significantly improved. It is now much clearer and also provides more support for the conclusions. It could nevertheless benefit from further clarification concerning two of the previously mentioned points (1,2). In addition, I am worried about two findings that are in clear contradiction to previous findings (3,4).

1 - I have to admit that I am still confused by the concept of the mileage and especially its novelty and benefits as well as its fundamental difference to the concept of life expectancy. This requires more clarification. In particular, in the response (page 9) it is mentioned that "the bond mileage calculated in our work is in fact from the actual number of strains that different bonds have endured under shear". Does this imply that the mileage is just the total strain?

In general, some of the terminology and explanations could still benefit from clarification. For example, it is not clear to me why the shear-thinning regime ($0.01 < Mn < 1$) is called the "shear rejuvenation regime" (page 3). In what sense and in what extent are the processes in this regime linked to what usually is considered as shear rejuvenation?

In addition, the meaning of the parameters shown in Fig. 4d (main figure) and their differences should be explained more clearly.

2 - The discussion of the usefulness of the Mason number has been considerably strengthened by adding a second attraction strength. For some parameters the Mason number indeed seems to be a very good choice. However, for a few parameters this is not the case. In Fig. 2b different behaviour is shown, as is in Fig. 3c. Concerning the latter, on page 6 this is attributed to the higher shear rate required for the stronger gel to achieve the same Mn. Does this not imply that, at least in this case, the shear rate is the crucial parameter and not the Mn? Furthermore, would the data fall on top of each other if plotted versus shear rate (for high shear rates)?

3 - The transient behaviour shown in the inset to Fig. 4d shows the first maximum between about 1 and 6. This is significantly higher than in other studies, e.g. the cited refs. 32, 39, 43 but also ref. 38, where the maximum was found between about 0.4 and 1.

4 - In the response to the question on the definition of the diffusion time (page 14 of the response) the mean-squared displacement (MSD) of a single colloid is shown as a function of time t . As given in the figure, one of the most fundamental results of colloid physics is $MSD = 6 D t$. This, however, is not at all what the figure shows. The slope in the log-log plot is neither one nor is it constant. Since this result was obtained in the most simple scenario, much simpler than in the remainder of the manuscript, this is very worrying. Something must have gone utterly wrong with the simulations and with the analysis (how was $D = 0.096$ obtained?). This does not boost my confidence in the results.

5 - Minor points.

Fig. 1d-g and Fig. 2a. The value of U_0 should be mentioned.

Fig. 2b is discussed before Fig. 2a.

Fig. 3b. What are the units of τ_B ? I guess τ_0 but this should be mentioned.

The section giving the simulation parameters should be updated. In particular, the new value of the attraction strength should be mentioned.

Response to comments by Reviewer A:

Reviewer 1 (Remarks to the Author):

I am satisfied with the response of the authors to my comments and also found their response to the comments of the other referee highly satisfactory. I believe this is an important paper that will be of great interest to a broad audience, and am happy to recommend it for publication in Nature Communications in its present form.

We are grateful and pleased to see the final recommendation of the referee one.

Response to comments by Reviewer B:

Reviewer 2 (Remarks to the Author):

The manuscript has been significantly improved. It is now much clearer and also provides more support for the conclusions. It could nevertheless benefit from further clarification concerning two of the previously mentioned points (1,2). In addition, I am worried about two findings that are in clear contradiction to previous findings (3,4).

We would like to thank the referee one more time for bringing these points to our attention. We have further made improvements/edits to the manuscript based on the comments by the referee, and believe that the current version satisfactorily addresses all concerns raised by the referee.

Q-1 - I have to admit that I am still confused by the concept of the mileage and especially its novelty and benefits as well as its fundamental difference to the concept of life expectancy. This requires more clarification. In particular, in the response (page 9) it is mentioned that “the bond mileage calculated in our work is in fact from the actual number of strains that different bonds have endured under shear”. Does this imply that the mileage is just the total strain?

A-1 - There are two parts to this question, and perhaps we haven't explained the concept clearly in the response letter previously submitted. With regards to the first part which concerns the differences between the bond mileage and shear strain: One should note that strain is inherently a flow measure, and not a bond measure. So, we intentionally wanted to differentiate the two, as what we refer to and what in fact matters in terms of source of stresses is the bond information, and not necessarily flow characteristics. For instance, for a single bond that exists at the very start of the shearing protocol, bond mileage equates the total strain at the time of the bond's rupture, yet for bonds that form under flow (which constitute majority of the bonds studied here), the bond mileage does not equate the total strain: a bond that is formed after γ strains and ruptured at $\gamma + \delta$ strains, bond mileage of this specific lifetime of this bond is calculated as $\gamma_{Bond} = \delta \neq \gamma + \delta$. Since there is a constant rupture/formation of bonds at different points during flow, presenting the life of bonds in terms of strain inherently makes it confusing for the reader, and thus we chose the bond mileage as our measure. In this definition a $\gamma_{Bond} = 1$ refers to the entire population of bonds that formed and ruptured after one strain during steady state flow, but a $\gamma = 1$ refers to one strain unit of flow after the start up. On the second point which concerns the differences between bond lifetime and bond mileage: first and foremost, the life expectancy of the bonds is a measure of time, with units of seconds (here, normalized by the diffusive time), while the bond mileage is a measure of deformation and thus dimensionless. The life

expectancy under a given deformation rate gives the most likely life time of a bond, and thus is strictly a rate-dependent measure. This doesn't mean that all bonds will live that long, but instead is the characteristic time of bond rupture for that given flow rate. One can multiply that by the shear rate and define a characteristic bond mileage, but this is not what we have done in the manuscript, since this average bond mileage will not provide any meaningful information about stresses borne in the system. Instead, the bond mileage as described above comes directly from the population of bonds that form at different times of flow, and rupture after a given strain unit. As such this measure is not rate dependent. Thus we stress that both concepts are essential in better understanding the dynamics of the system and should be used in conjunction.

Page 6-“... The bond mileages presented here are sampled over the entire range of quasi steady-state flow. To do this, all populations of bond life times and their corresponding stresses are tracked and data are collected over the entire flow protocol. Note that although bond mileage is of the same nature as shear strain, it reflects on a population of bonds that share the same total number of flow strains, regardless of the point they were formed. Thus it is important to differentiate between the bond mileage and shear strain, as one refers to all bonds that are formed [at different times] and ruptured after a certain number of strains, and the other measures the flow strains.”

In general, some of the terminology and explanations could still benefit from clarification. For example, it is not clear to me why the shear-thinning regime ($0.01 < Mn < 1$) is called the “shear rejuvenation regime” (page 3). In what sense and in what extent are the processes in this regime linked to what usually is considered as shear rejuvenation? In addition, the meaning of the parameters shown in Fig. 4d (main figure) and their differences should be explained more clearly.

A-1 - The reason the term “shear rejuvenation” is used for $0.01 < Mn < 1$ is that in this regime, gel microstructure is not completely fluidized and the particles still form a microstructure, however this microstructure is entirely different from the one at quiescent conditions or under shear at first regime $Mn < 0.01$, hence this regime is called “rejuvenation regime”. This is consistent with a series of previous reports we have published where we focus on these different flow regimes (Jamali, McKinley and Armstrong, PRL 2017, Mat. Tod. Adv. 2020, PRL 2019). We have made a clarifying statement in the manuscript on this as well as the definition of all parameters in the caption of figure four.

Page 3-“... In the first regime, the colloidal network is rather unchanged with slight coarsening and compaction over time, and in the fluidization regime the network is nonexistent as the colloidal bonds are effectively broken. In the second regime secondary flow-induced structures emerge that are entirely different from the initial gel structures, and thus we refer to this regime as shear-rejuvenation. It should be noted that some literature refer to shear-rejuvenation when a high rate of shear is applied to remove any previous memory, followed by cessation of flow to re-construct the gel microstructure.”

Caption for Figure 4-“... **a)** The contribution of the attractive forces, $\sigma_{Att}(\gamma_B)$, **(b)** the contribution of hydrodynamic forces, $\sigma_{Hyd}(\gamma_B)$, and **(c)** the total stress, $\sigma_{Tot}(\gamma_B)$, measured for different bond populations, represented as the total bond mileage, and normalized by the total stress measured over all bond mileages in the system for different Mason numbers. **(d)** The characteristic bond mileage from life expectancy, $(\gamma_B^* = \tau_0\dot{\gamma})$, the critical strain for stress overshoot under flow start up experiments (γ_c) , and the bond mileage with the maximum attractive stresses (γ_{BC}) against the imposed Mn number. The insert graph shows the stress against strain curves upon inception of shear flow, from which the critical strain for stress overshoot is measured for a few sampled Mason values.”

Q-2 - The discussion of the usefulness of the Mason number has been considerably strengthened by adding a second attraction strength. For some parameters the Mason number indeed seems to be a very good choice. However, for a few parameters this is not the case. In Fig. 2b different behaviour is shown, as is in Fig. 3c. Concerning the latter, on page 6 this is attributed to the higher shear rate required for the stronger gel to achieve the same Mn. Does this not imply that, at least in this case, the shear rate is the crucial parameter and not the Mn? Furthermore, would the data fall on top of each other if plotted versus shear rate (for high shear rates)?

A-2 - Here we use Mason number because it can explain the overall bulk response of the gel to deformation, yet we don't expect (and we have no claim that) Mn number to single-handedly categorize the entire observed microstructural features (Fig. 2b) or the kinematics of bond rupture/formation (Fig. 3c). The reviewer is absolutely correct about the shear rate controlled nature of bond kinematics at high shear rates, in fluidized regime, because of the high deformation rate, and relatively small number of bonds, the kinematics of bond rupture/formation is a rather collision rate controlled phenomenon, which means shear rate is indeed the crucial parameter for that regime. To further accentuate this point in the manuscript, we have added an insertion to Fig. 3c, where we plot the bond rupture/formation frequency versus shear rate (Fig.), it's clear that the data for different attraction levels indeed fall on top of each other in that fluidized regime. We have also modified the manuscript to further accentuate this point. One can think about this regime as: in the fluidized regime there aren't as many colloidal bonds to begin with, and thus the likelihood of bond formation and rupture depends on the colloisional dynamics and relative velocity of particles under flow, which is reflected through shear rate rather than Mason number. But in other regimes, where there bonds resisting to flow at different velocities as imposed by flow, Mason becomes the significant indicator for the dynamics.

Page 5-“... . For higher Mn, the rupture rate of bonds for the stronger gel ($D_0 = 20kT$), become more frequent than that of the weak gel ($D_0 = 6kT$). This is expected, as a higher shear rate is required for the stronger gel, to achieve the same Mn , compared to the weak gel. Thus at high deformation rates, the bond dynamics are controlled/described through shear rate rather than the Mason number (inset of Fig. 3c shows the average bond rupture/formation frequency of the two gels fall on top of each other when plotted against shear rate). In this regime, since shear flow effectively breaks down majority of colloidal bonds,

these rates are controlled by the collisional dynamics and the relative velocity of particles, i.e. shear rate; however, at lower deformation rates and since colloidal bonds resist motion in different planes of shear, Mason number becomes the key indicator of these dynamics.”

Q-3 - The transient behaviour shown in the inset to Fig. 4d shows the first maximum between about 1 and 6. This is significantly higher than in other studies, e.g. the cited refs. 32, 39, 43 but also ref. 38, where the maximum was found between about 0.4 and 1.

A-3 - It is quite common in computational studies for the first maximum of the stress overshoot depending on different simulation parameters such as the thermostat temperature ($k_B T$), dissipative coefficient (γ), shearing protocol etc.. Nonetheless and despite the difference between our reported results and other studies, we still observe a similar scaling with previous works¹. We have modified the manuscript to ensure that the difference between our reported results and the previous works is clearly communicated to the readers. Nonetheless, we would like to assure the referee that while these quantitative differences exist, the scaling behavior discussed is similar in all these studies. One of our own previous reports, Boromand et al. *Soft Matter* 2017, also reports strains in the 0.4-1.5 regime, consistent with the experimental literature, using the same simulation platform with different parameters. This further strengthens our confidence in that the scaling behavior will remain unchanged by changing the details of simulation parameters.

Page 5-“... The inset graph in Fig.4.d shows the stress overshoot at the initial shear start-up flow. our results suggest that these critical strains at which stress overshoot is observed, γ_C , also show the same trend and scaling as the maximum in the stress-bond mileage graphs. This is significant, since the bond mileages are all calculated in the quasi-steady state and far from this initial departure from the linear viscoelastic response, and yet reflect on the microstructural and dynamical sources of the stress overshoot. It’s worth mentioning that our simulations predict higher strain values at the stress overshoot compared to other works¹⁻⁵, while the same scaling is observed. This difference arises from the dependency of the critical strain on the simulation parameters and the shearing protocol, therefore, our discussion is solely on the scaling behavior for the critical strain at stress overshoot, γ_C , the critical bond mileage at attractive stress overshoot, γ_{BC} , and the critical expected bond mileage, γ_B^* ...”

Q-4 - In the response to the question on the definition of the diffusion time (page 14 of the response) the mean-squared displacement (MSD) of a single colloid is shown as a function of time t . As given in the figure, one of the most fundamental results of colloid physics is $MSD = 6Dt$. This, however, is not at all what the figure shows. The slope in the log-log plot is neither one nor is it constant. Since this result was obtained in the most simple scenario, much simpler than in the remainder of the manuscript, this is very worrying. Something must have gone utterly wrong with the simulations and with the analysis (how was $D = 0.096$ obtained?). This does not boost my confidence in the results.

A-4 - The reviewer is absolutely right about the erroneous values in the MSD plot, and we have to admit this was result of a rushed simulation in response to the review. However, we would like to reassure the referee that the simulations have been performed very carefully and the value of the diffusive time/diffusion coefficient $D = 0.096$ had been calculated months ago and from the correct plots, now made here. Clearly, there has been a clear mistake in generating this figure: the very initial stage of simulations in DPD are very widely known to recover a ballistic motion (before thermal equilibrium), followed by a clear transition to diffusive motion at longer times. Unfortunately, we had made the previous figure only on the initial part, which has a slope of $\sim 1 - 2$ and also changing with time as the referee pointed out. The complete graph, including long time diffusive behavior and the way we calculate the diffusion coefficient are now clearly indicated in the following figure:

Q-5 - Minor points.

Q-5.1 - Fig. 1d-g and Fig. 2a. The value of U_0 should be mentioned.

A-5.1 - We have modified the caption of Fig. 1d in the manuscript to include the attraction strength of the shown particles:

Page 5-“Shear stresses, calculated at quasi-steady state, plotted against Mason number, Mn , in the form of: (a) fluid, colloidal, and total stresses, (b) the relative contribution of each phase’s stresses to the total stress, and (c) the relative contribution of the attractive and hydrodynamic microscopic forces to

the macroscopic stress. The filled symbols and solid lines correspond to attraction strength of $U_0 = 6k_B T$, and open symbols with dashed lines correspond to attraction strength of $U_0 = 20k_B T$ between colloids. Snapshots of the colloidal particles, color coded with contact number at: **(d)** quiescent condition, $Mn = 0.0$, and the shear-induced structuration regime: **(e)** $Mn = 0.03$, **(f)** $Mn = 0.24$ and, **(g)** $Mn = 0.7$. All snapshots correspond to attraction strength of $6k_B T$ ”

Q-5.2 - Fig. 2b is discussed before Fig. 2a.

We have swapped panel placements of Fig. 2 and also modified the caption of Fig 2 in the manuscript to indicate that the data shown in Fig. 2a belongs to the $6k_B T$ strength of attraction.

Page 5-“(a) The average contact number at each Mason number at the quasi-steady state. Values are presented for two different attractions strengths of $U_0 = 6, 20k_B T$, with the initial value of the average coordination number marked with a dashed horizontal line. The insert graph show the normalized coordination number by its initial value [of a quiescent gel]. And (b) distribution of contact numbers (dashed line) and the relative contribution of stresses borne on each contact population (solid lines) against coordination number for different imposed deformation rates. The insert shows the contribution of stress per particle, for each coordination population ($U_0 = 6k_B T$).”

We have also modified the manuscript to make sure that the correct panel is discussed/referred to, and those are in the correct order as well.

Page 5-“... . Using this criteria, the coordination number of colloidal particles and their distributions are quantified and represented in Fig. 2. The average coordination numbers, $\langle Z \rangle$ as a function of applied Mason number, plotted in Fig. 2.a, is in agreement with previous experimental measurement of ...”

Page 5-“... . In the initial shear compaction regime, the average value of the coordination number for the entire system grows slightly above its measure at the quiescent conditions, as indicated in the insert graph of Fig. 2.a. However, the extent of this increase depends upon the volume fraction of particles, and is more visible for the lower fraction of colloids in the system.”

Q-5.3 - Fig. 3b. What are the units of τ_B ? I guess τ_0 but this should be mentioned.

A-5.3 - The units of τ_B are indeed τ_0 , and we have modified the caption of Fig. 3b accordingly:

Page 5-“a The relative distribution of bond life times normalized by the diffusion time of a bare particle for different Mn numbers. Dashed lines represent an exponential decay fitting function as $P(\tau) = C \exp(-\tau/\tau_B)$, where C is a conversion factor constant. The insert graph shows the same probabilities in a semi-log graph. b the life expectancy (in τ_0 units) calculated from the life time

distributions versus Mn number, with the insert graph showing the same data in a log-log plot. and \mathbf{c} the average number of formed/ruptured bonds per particle during a single diffusion time. The filled symbols and solid lines correspond to attraction strength of $U_0 = 6k_B T$, and open symbols with dashed lines correspond to attraction strength of $U_0 = 20k_B T$ between colloids.”

The section giving the simulation parameters should be updated. In particular, the new value of the attraction strength should be mentioned.

We have added the $20k_B T$ attraction to the simulation parameter section as follows:

Page 5-“...Prior to imposing shear flow, colloidal particles with volume fraction of ($\phi = 0.2$) form space-spanning gel networks. The simulation box includes 518,400 solvent particles and 10,313 colloidal particles where simulation box size is 60 times the particle radius ($a = 1$) in all directions, number density of ($\rho = 3$) is used for the solvent particles at dimensionless temperature of ($k_B T = 0.1$). Densities are matched by setting the mass of a colloidal particle to ($m_C = 4/3\rho\pi a^3$), with the solvent particles of unit mass ($m_S = 1.0$). The strength of attraction potential is set to $U_0 = 6k_B T$ for the weak gel, and $20k_B T$ for the strong gel with $\kappa = 33$ acting over a range of $0.1a$ to achieve a short-ranged weak attraction. Once colloidal gels are prepared, a constant shearing protocol is applied in a Mason range of $0.003 < Mn < 3.5$.”

References

- [1] Arman Boromand, Safa Jamali, and João M Maia. Structural fingerprints of yielding mechanisms in attractive colloidal gels. *Soft matter*, 13(2):458–473, 2017.
- [2] Jader Colombo and Emanuela Del Gado. Stress localization, stiffening, and yielding in a model colloidal gel. *Journal of rheology*, 58(5):1089–1116, 2014.
- [3] Esmaeel Moghimi and George Petekidis. Mechanisms of two-step yielding in attractive colloidal glasses. *Journal of Rheology*, 64(5):1209–1225, 2020.
- [4] N Koumakis and G Petekidis. Two step yielding in attractive colloids: transition from gels to attractive glasses. *Soft Matter*, 7(6):2456–2470, 2011.
- [5] Esmaeel Moghimi, Alan R Jacob, and George Petekidis. Residual stresses in colloidal gels. *Soft Matter*, 13(43):7824–7833, 2017.

REVIEWERS' COMMENTS

Reviewer #2 (Remarks to the Author):

The authors clarified all the points I raised and improved the manuscript significantly to which I congratulate them. I am now very happy to recommend the manuscript for publication in Nature Communications. (With one tiny suggestion. In the caption to figure 3b the units should be mentioned, as proposed in the rebuttal, i.e. '... the life expectancy (in τ_0 units) ...'.)